# CA1 cell activity sequences emerge after reorganization of network correlation structure during associative learning

**Mehrab N Modi, Ashesh K Dhawale†‡, Upinder S Bhalla\***

National Centre for Biological Sciences, Tata Institute of Fundamental Research, Bangalore, India

**Abstract** Animals can learn causal relationships between pairs of stimuli separated in time and this ability depends on the hippocampus. Such learning is believed to emerge from alterations in network connectivity, but large-scale connectivity is difficult to measure directly, especially during learning. Here, we show that area CA1 cells converge to time-locked firing sequences that bridge the two stimuli paired during training, and this phenomenon is coupled to a reorganization of network correlations. Using two-photon calcium imaging of mouse hippocampal neurons we find that co-time-tuned neurons exhibit enhanced spontaneous activity correlations that increase just prior to learning. While time-tuned cells are not spatially organized, spontaneously correlated cells do fall into distinct spatial clusters that change as a result of learning. We propose that the spatial re-organization of correlation clusters reflects global network connectivity changes that are responsible for the emergence of the sequentially-timed activity of cell-groups underlying the learned behavior.

**\*For correspondence:**
bhalla@ncbs.res.in

**Present address:** †Department of Organismic and Evolutionary Biology, Harvard University, Cambridge, United States; ‡Center for Brain Science, Harvard University, Cambridge, United States

**Competing interests:** The authors declare that no competing interests exist.

## Introduction

The mechanisms of memory formation have been the subject of considerable study (*Morris et al., 1988*; *Kandel, 2001*). Much evidence points to Hebbian plasticity as the neural mechanism for the association of two co-occurring stimuli (*Bliss and Collingridge, 1993*; *Morris, 2003*). However, this mechanism alone is not sufficient to account for learning under conditions where the two stimuli are separated in time by more than 100 ms (*Levy and Steward, 1983*), as has been commonly observed (*Solomon et al., 1986*; *Baeg et al., 2003*).

One example of such a time-bridging task is trace eyeblink conditioning, where the goal is to associate a neutral tone or conditioned stimulus (CS) with a temporally separated, aversive puff of air to the eye, or unconditioned stimulus (US) (*McEchron and Disterhoft, 1999*). After many pairings of these stimuli, the subject learns to blink in response to the tone, even though the tone and puff never overlap in time (*Tseng et al., 2004*; *Kalmbach et al., 2009*). Lesion studies have shown that the hippocampus is required for learning the trace conditioning task, but not a related delay conditioning task, where the CS and US overlap in time (*Büchel et al., 1999*; *Tseng et al., 2004*). These observations indicate a role for the hippocampus during the association of temporally discontiguous events (*Wallenstein et al., 1998*), specifically during the 'trace' period separating stimulus pairs.

How might a network of neurons maintain a representation of the stimulus through time? Two possible models have been proposed. The first model hypothesizes that the representation of the first stimulus is maintained by the sustained firing of stimulus-selective cells through a trace interval (*Solomon et al., 1986*). Such a model is supported by observations of sustained firing by neurons in the medial prefrontal cortex (*Fuster, 1973*; *Baeg et al., 2003*) and the medial entorhinal cortex (*Egorov et al., 2002*) during working memory tasks. An alternative model proposes that sensory representations are maintained by the sequential activation of groups of neurons (*Levy et al., 2005*;

**eLife digest** Ivan Pavlov famously discovered that dogs would salivate upon hearing a bell that had previously been used to signal food, even when there was no food present. This ability to connect events that occur close together in time is known as associative learning. But how is it supported within the brain?

In the late 1940s, neuroscientist Donald Hebb proposed that if one neuron persistently and repeatedly takes part in firing a second neuron, the connection between the two neurons will be strengthened. Thus, if neurons that encode the sound of a bell are active at the same time as neurons that encode receiving food, connections between the two groups will be strengthened, and this might enable the dogs to associate the two events.

However, animals can also learn to associate events that do not overlap in time. For example, we can associate a bout of food poisoning with a meal we consumed several hours earlier. In rodents, this type of learning is often studied using a task known as trace eyeblink conditioning, in which a tone signals the delivery of a puff of air to the eye after a short delay. Rodents eventually begin to blink in response to the tone, even thought the tone and the air puff are never presented simultaneously.

Two possibilities have been proposed for how this might occur: either the neurons that encode the tone remain active until delivery of the air puff, or different groups of neurons are successively activated in a relay that spans the interval between the tone and the air puff. Now, Modi et al. have used in vivo imaging in awake mice to obtain evidence in favour of the second option.

Mice were trained on the conditioning task while imaging was used to follow the activity of neurons in a region of the brain known as the hippocampus. As animals learned the task, neurons in part of the hippocampus called CA1 began to reorganize their firing patterns so that distinct groups of cells were active at each time point in the interval between the tone and the air puff. By contrast, hardly any neurons were active across the entire delay. The organized firing became particularly apparent at the same time as the mice first began to blink in response to the tone, and was only ever seen in animals that learned the task successfully.

As well as providing evidence to distinguish between competing theories of associative learning across a delay, this study is the first to follow in real-time the reorganization of networks of neurons within the hippocampus during this common type of learning.

*Howe and Levy, 2007*; *MacDonald et al., 2013*). This view arose from modeling studies that trained simple hippocampal area CA3 network models on the trace conditioning task. Rather than observing sustained firing, the authors found that groups of neurons began to show activity in well-timed, sequential bouts. Neurons representing the CS kicked off this 'relay' of activation, which eventually activated US representing neurons at the appropriate time (*Levy et al., 2005*; *Howe and Levy, 2007*). However, this model awaits experimental verification.

Sequential activity in hippocampal CA1 cells has been seen previously, albeit in some very different behavioral contexts, including temporal memory tasks (*Louie and Wilson, 2001*; *Pastalkova et al., 2008*; *Gill et al., 2011*). There has also been a series of recent, more closely related studies of hippocampal activity, in which rats or monkeys performed stimulus-retention tasks where they had to remember an odor or visual stimulus in order to receive a reward. Here too, hippocampal CA1 neurons were observed to be active in stimulus-triggered, time-locked sequences (*MacDonald et al., 2011*; *Naya and Suzuki, 2011*; *Kraus et al., 2013*; *MacDonald et al., 2013*). It is increasingly clear that hippocampal CA1 cells adopt sequential activity patterns when subjects are placed in a behavioral context requiring the bridging of temporally separated stimuli. However, in all the studies where this has been observed, only well-trained subjects were used, leaving the time course and mechanism of the emergence of sequentially timed activity entirely unknown.

Despite sequential activity having been implicated in several temporal memory tasks, there is little experimental data on the network changes that underlie its emergence. Functional connectivity, as measured by correlations between neuronal activity in the absence of stimulus presentation, is one way to monitor such network changes (*Ts'o et al., 1986*; *Bair et al., 2001*; *Fujisawa et al., 2008*). In a study using two photon calcium imaging of motor cortex pyramidal neuron activity, changes

in spontaneous activity correlations have been inferred to indicate learning-related circuit plasticity (*Komiyama et al., 2010*). In another study where rats were exploring a novel track, correlations between pairs of place cells increased with increasing exposure to the novel environment (*Cheng and Frank, 2008*; *Dragoi and Tonegawa, 2013*). With the ability to measure changes in input from upstream circuits using spontaneous activity correlations across many cells in the network, two-photon recordings allow the testing of predictions from the model proposed for the emergence of sequential activity (*Levy et al., 2005*; *Howe and Levy, 2007*). Furthermore, such recordings provide relative cell locations within the hippocampus, allowing one to examine the spatial organization of activity patterns (*Hampson et al., 1999*; *Brivanlou et al., 2004*; *Kjelstrup et al., 2008*). As learning progresses, one should see changes in spontaneous correlations reflecting altered inputs and changes in network connectivity.

Two key questions remain unanswered in the absence of data recorded from large numbers of neurons during training on a temporal memory task – how do sequential activity representations in the hippocampus emerge during learning, and what are the underlying changes in network connectivity? In this study, we train mice on a trace eyeblink conditioning task while recording activity from populations of area CA1 neurons using two photon calcium imaging. In order to eliminate the influence of running or changing spatial position on hippocampal activity, we implemented a trace eye-blink conditioning task for head-fixed mice. Further, our activity measurements revealed CA1 network dynamics during learning, as we began with naïve mice and trained them to criterion within the recording session. We found that sequentially timed activity of groups of area CA1 cells emerged progressively during the course of learning. Additionally, mean spontaneous activity correlations at the network level rose transiently, while only the correlations between co-tuned cells remained elevated towards the end of the session. Finally, we observed spatially organized clusters of neurons that had elevated spontaneous activity correlations. These correlation clusters re-organized during learning.

## Results

Using two-photon calcium imaging, we monitored the activity of large numbers of neurons in area CA1 of the dorsal hippocampus in awake, head-fixed mice while they were trained on a trace eyeblink conditioning task (*Figure 1A*).

### Mice learn a trace eyeblink conditioning task within a single session

Head-restrained mice were trained on a trace eyeblink conditioning task (*Tseng et al., 2004*), where they learned to associate a neutral tone stimulus (Conditioned Stimulus–CS) with an aversive puff of air to the eye (Unconditioned Stimulus–US) within a single session ('Materials and methods-Behavioral training'). Tone and puff were non-overlapping and separated by a 250 ms interval, thus requiring the subject to maintain a representation or 'trace' of the tone (CS) in order to associate it with the puff (US). Behavioral responses were measured by recording deflections in eyelid position (*Figure 1B*).

We found that naïve mice responded to tone presentation with small, but distinct and measurable eyelid movements early in training, even on trials prior to the introduction of the puff stimulus (*Figure 1C*). Following repeated pairings of tone and puff, however, blink responses to tone increased significantly (>2 standard deviations [SD]) in amplitude and duration, ($n$ = 9 of 18 mice, *Figure 1B,D*). Conditioned Response (CR) trials were defined as those trials in the training session whose area under the eyelid–position curve, during the interval spanning tone onset to puff onset, was significantly larger than the pre-training baseline ('Materials and methods-Behavioral training'). We next established a performance score by measuring the ratio of CR blink rates to spontaneous blink rates ('Materials and methods-Data analysis', *Figure 1E,F*). As a learning control, we randomized the relative timing of tone and puff from trial to trial, for a different set of mice. These pseudo-conditioned mice did not demonstrate an increase in blink amplitude or duration (performance scores, trace = 8.74 ± 2.73, pseudo = 1.96 ± 0.44, mean ± standard error of the mean (SEM); *Figure 1E,F*, two-sample $t$ test, p=0.012). 9 of 18 trace conditioned mice had performance scores higher than those of the pseudo-conditioned mice.

We next examined the performance of each mouse, to assess which individual mice learned the task, and if so, at what trial number in the training session. We used a previously described expectation maximization algorithm to assess whether each individual mouse had learned the association, and to obtain learning curves (*Smith et al., 2004*). Briefly, the algorithm uses the list of CR trials for a given mouse, along with the chance probability of the occurrence of a well-timed, significant blink to estimate the probability of CR production at each trial in the session (*Figure 1G*). Furthermore, the

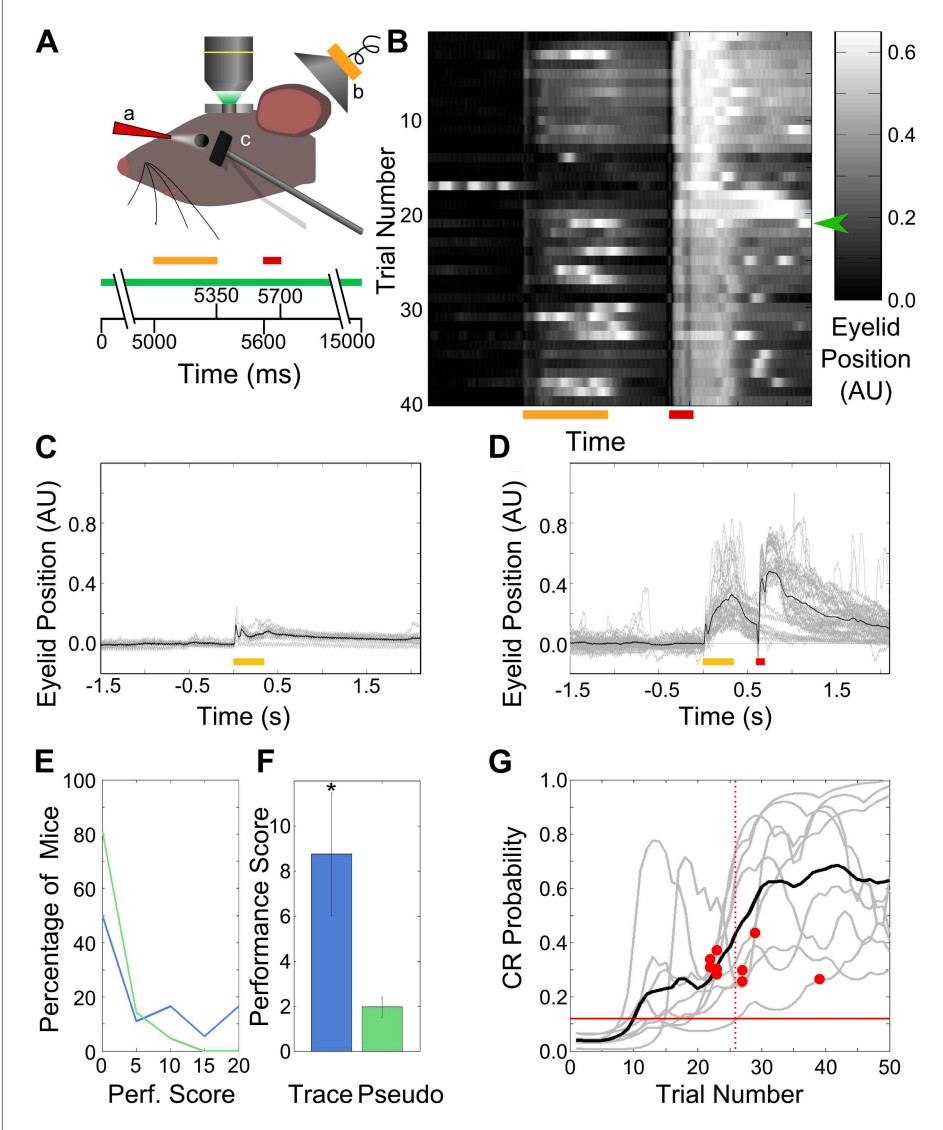

**Figure 1**. Behavior: trace eye-blink conditioning of mice. (**A**) Cartoon schematic of experimental system. Two-photon calcium imaging was carried out in area CA1 of the dorsal hippocampus in a head-fixed mouse which underwent trace eyeblink conditioning. A speaker (yellow, b) was used for tone stimulus delivery, while a nozzle (red, a) directed the aversive air-puff. A magnetometer (black, c) was used to monitor eyelid position in order to detect blinks. Scale bars at the bottom of the figure indicate times of stimulus delivery (yellow and red bars for tone [350 ms long] and puff [100 ms long] respectively, along with the gap in-between [250 ms]) as well as data acquisition (green bar, 15 s long) during a single trial. (**B**) Sample eyelid position signal traces from a mouse undergoing trace eyeblink conditioning. The color scale indicates eyelid position in arbitrary units, with high values indicating eyelid closure (blink). This mouse started reproducibly showing significant blinks before the air-puff (e.g., green arrow) mid-way through the session. (**C** and **D**) Eyelid position traces in response to pre-training tone presentation (**C**), and both tone and puff stimuli during trace eye-blink conditioning (**D**). Gray traces are from individual trials and black traces are averages across trials. Yellow and red bars at the bottom indicate times of delivery of tone and air-puff respectively. (**E**) Distributions of performance scores for trace (blue) and pseudo (green) conditioned mice. (**F**) Average performance score, which is the ratio of tone evoked significant blink (CR) rates to spontaneous blink rates, is plotted for all trace conditioned mice (blue) and pseudo-conditioned mice (green). Error bars indicate SEM. * indicates p<0.05. (**G**) Learning curves (gray lines) for the nine mice that learned the association to criterion. The learning trial identified for each mouse is marked with a red circle. The black curve shows average performance across mice. The vertical, dotted line indicates the mean learning trial across mice (trial 26). Each learning curve was obtained by first obtaining a binary list of significant response trials for each mouse, and then using a previously described expectation maximization algorithm to calculate the CR probability on each trial, for individual mice. The horizontal, red line indicates the probability of CRs by chance.

trial at which a mouse has learned the task is statistically defined. It is the first trial when the lower 95% confidence interval of the probability of CR production rises and remains above chance. As per this criterion, 9 out of 18 trace conditioned mice learned the association. The mean of the individual learning trials was 26 ± 5 (mean ± SD, n = 9 learners; for six of these learner mice, imaging data was also acquired, *Video 1* shows high-speed video of mouse blinks before and after learning). These were the same nine mice that also had higher performance scores than pseudo-conditioned controls.

Half the mice trained on the trace eyeblink conditioning task failed to learn to criterion. In part, this was because we were restricted to training mice for only a single session due to the limited residence time of calcium indicator dye in cells (*Stosiek et al., 2003*). These non-learners most likely represent a heterogeneous population of mice at different stages of learning involving multiple brain regions (*Kalmbach et al., 2009*). In other words, given more training sessions, many of these mice would likely have learned this task to criterion. Consequently, the interpretation of area CA1 calcium imaging data from 'non-learners' is complicated by the uncertain state of learning of each mouse. Hence, in most analyses, we have not used data from these mice. However, for completeness, we have included data from non-learners in key figures.

## Cells in hippocampal area CA1 imaged from awake, behaving mice exhibit temporal tuning

We surgically exposed the left, dorsal hippocampus of naïve mice and bolus loaded a synthetic calcium indicator dye. We then implanted cranial windows through which we imaged calcium responses from cells in area CA1 of the hippocampus (*Figure 2A*, *Figure 2—figure supplement 1A*, *Video 2* shows calcium responses from a sample field of view). Image acquisition was carried out within a field of view covering 96 ± 29 (mean ± SD) cells per mouse, imaged at frame rates ranging from 11 Hz to 16 Hz. In parallel with calcium imaging, we simultaneously measured eyeblink responses of mice over the entire pre-training and training sections of the conditioning protocol (n = 14 trace conditioned and six pseudo-conditioned mice; 'Materials and methods-Awake, two photon calcium imaging of area CA1 cells'; *Figure 2—figure supplement 1B*). Cell bodies were imaged from the visually identified *stratum pyramidale* layer of the hippocampus (*Figure 2—figure supplement 1C*), at depths ranging between 135–150 μm below the hippocampal surface (supporting *Video 2* shows imaged calcium responses and *Video 3* shows a stack of optical, z-section images with the densely labeled cell-body layer visible). To ensure the reliability of our calcium fluorescence data, we carried out two checks. First, the frequency of calcium transients was observed to be 1.3 ± 0.2 Hz (mean ± SD). While this need not accurately reflect spike rates in our experimental system, it falls well within the range of spontaneous spike rates previously observed in CA1 cells, and was un-changed between early and late trials in the session (*Figure 2—figure supplement 1D*; *Czurkó et al., 1999*). Second, the summed area under the calcium curve was also calculated for all datasets, and those datasets that showed a significant drift between early and late trials were discarded (n = 1 of 21; *Figure 2—figure supplement 1E*).

We first looked for evidence of sustained or sequentially timed neuronal activity in calcium fluorescence traces for individual cells (ΔF/F,'Materials and methods: Data analysis'), on trials following the CR learning peak. First, we determined whether or not cells showed sustained activity through the entire period of interest by measuring how wide

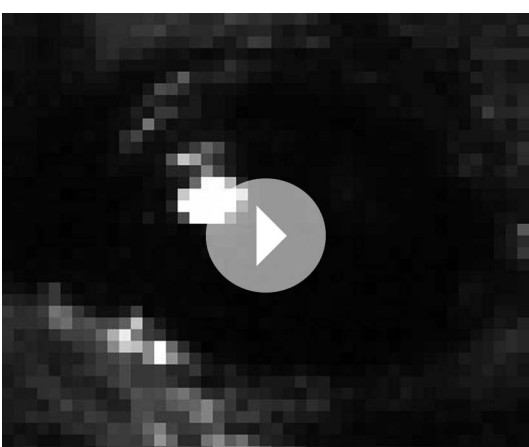

**Video 1**. Mouse Behavior. Video of mouse eyeblink behavior acquired at 100 frames per second (fps) and played back at 10 fps (i.e. 0.1x speed). A yellow spot in the bottom-right corner indicates when tone is being delivered, and a red spot indicates when the air-puff is being delivered. Frames from three trials, at different points in the session are shown, depicting behavior early in the session, prior to learning, and late in the session after learning. The last portion of the video shows eyeblink behavior in a probe-trial, where the tone was presented but no air-puff was delivered.

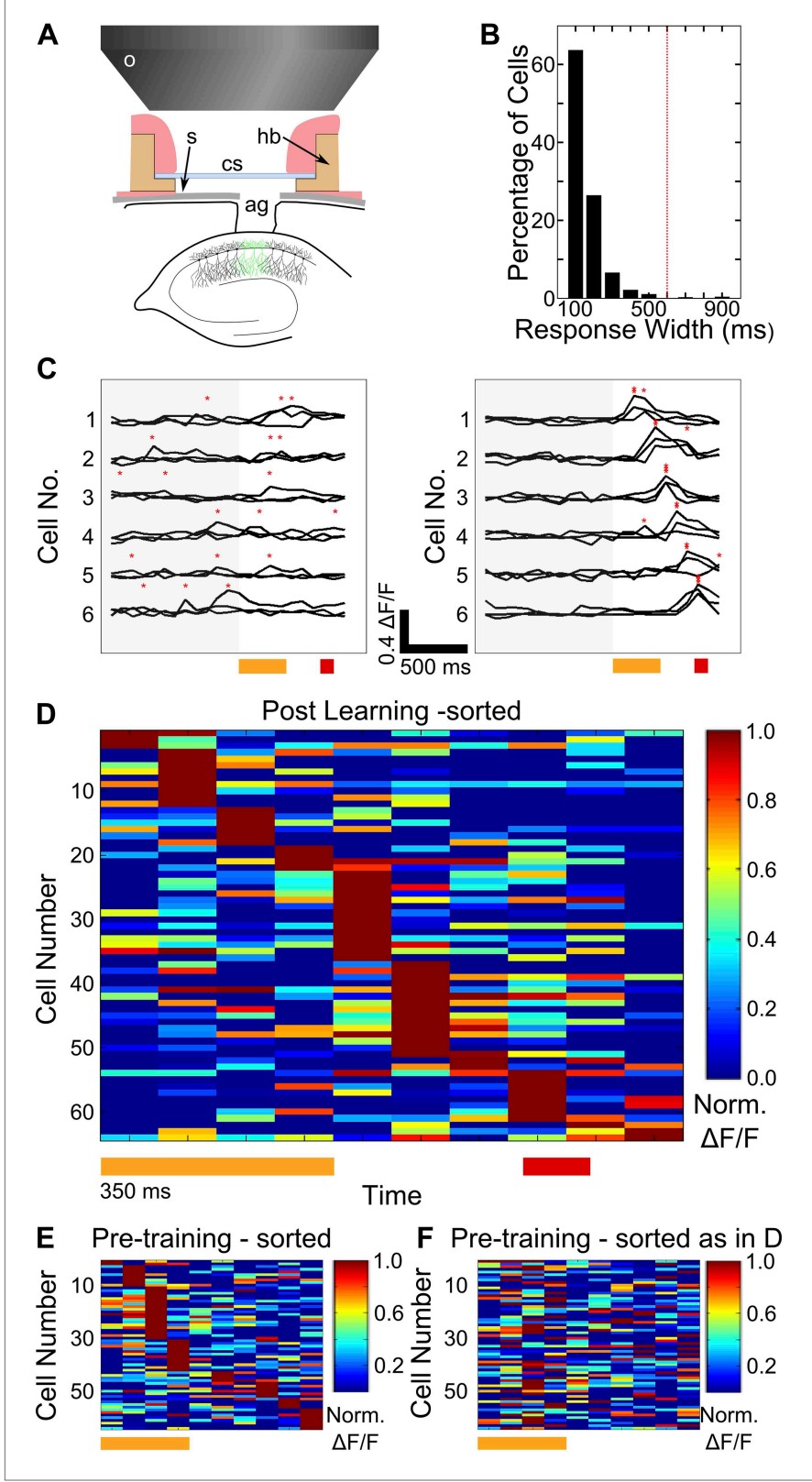

**Figure 2**. Two-photon imaging of calcium-responses in area CA1 neurons from awake mice. (**A**) Schematic of the imaging preparation. o–objective lens, s–skull, cs–cover slip, hb–head bar, ag–agarose. (**B**) Histogram of neuron response widths in ms, calculated as the time for which a given neuron's trial-averaged, ΔF/F trace remains above

*Figure 2. Continued on next page*

*Figure 2. Continued*

50% of the peak value. The red, dotted line indicates a response width of 600 ms, which is the time of interest between tone-onset and puff-onset. (**C**) Area CA1 cell responses show sequentially timed activity peaks after learning. Calcium response (ΔF/F) traces for six exemplar neurons from a single mouse, for sets of three trials before (panel on the left), and after (panel on the right) task learning. Neurons have been sorted as per the timing of the peak in the averaged trace. The yellow and red bars at the bottom represent the times of delivery of tone and puff respectively. The gray shading to the left covers the period of spontaneous activity prior to the onset of the tone. The red asterisks indicate the peak in each individual trace. Scale bars indicate 0.4 ΔF/F and 500 ms along the time axis. (**D**) Area CA1 cell activity peaks tile the entire CS-on to US-off interval. Area CA1 calcium response traces from an example dataset, sorted by the peak times of the responses. Each response trace has been averaged over all trials following the learning trial (*Figure 1G*), and has been normalized to the peak ΔF/F response value for each neuron. The yellow and red bars below indicate times of delivery of tone and air-puff respectively. 50% of the neurons from the field of view, with the most reliably timed responses have been shown. This is to make this plot comparable to the ones from subsequent analyses, where neurons have been similarly chosen. (**E** and **F**) Cell activity peak timings change during learning. In **E**, pseudo-colored ΔF/F traces for the period of interest during and after tone delivery, are plotted using data acquired during the *pre-training* session, where tones without air-puff were delivered. Cells have been sorted as per the timings of peaks in pre-training session data. The yellow bar at the bottom indicates time of delivery of tone (350 ms). In **F**, the same averaged activity traces as in **E** have been re-ordered according to each cell's activity peak timing *after* learning has occurred, as shown in (**D**) Plotted in *Figure 2—figure supplement 1*, are panels depicting the surgical preparation, the numbers of mice from each treatment group, images of dye-loaded tissue taken at multiple depths and basic data quality control analyses.

The following figure supplements are available for figure 2:

**Figure supplement 1**. Imaging preparation and calcium response data.

---

each cell's response was at half the amplitude of the peak of the response. The mean response width measured in this manner was found to be 120 ms ± 81 ms (mean ± SD, n = 542 cells, *Figure 2B*). Only three cells (~0.5%, of all cells) had response widths greater than 600 ms, the length of the period from tone-onset to puff-onset. Based on this analysis, we concluded that sustained cell activation is not the mechanism used by the hippocampus to maintain a stimulus representation. We next looked for indications of sequential activation. Prior to learning, calcium response peaks were not reliably timed across trials, relative to tone onset (*Figure 2C*, left panel). Post learning, however, we observed that area CA1 cells had activity peaks at fixed time-points relative to tone onset, seen consistently across multiple trials (*Figure 2C*, right panel). We further characterized this by averaging these ΔF/F activity traces over trials and identifying activity peaks within the time window from tone onset to 200 milliseconds after puff onset. We rank ordered cells based on the timings of their activity peaks and found that small groups of cells firing at each time-point, tiled the entire interval of interest (*Figure 2D*). When the same procedure (averaging followed by sorting by timing) was carried out on traces from the pre-training dataset, tiling was markedly skewed with most cells (66% in this case) showing activity peaks during the tone period (*Figure 2E*). Furthermore, if neurons in the pre-training dataset were ordered as per timing of peaks in the training session, no clear tiling was visible (*Figure 2F*). This indicated that timings of peak activity of area CA1 cells, relative to the onset of tone stimulus, changed during training. Furthermore, at the population level, these peak times appeared to tile the interval of interest between tone and puff.

## After learning, area CA1 cells show reliably timed, sequential calcium-responses

Having observed that area CA1 cells were active at fixed time-points relative to tone onset, we next wanted to quantify the reliability with which cells fired at these times. For each cell, we defined the timing of the peak in its averaged ΔF/F trace as its peak response time (PT). We then quantified the reliability with which cells fired at their respective PTs, from trial to trial (*Figure 3A* shows a cell reliably firing at a particular PT). First, we reasoned that if cells show time-aligned activity, the peak of the trial-averaged ΔF/F trace would be higher than if activity were not reliably timed. Hence, we computed the peaks of the averaged ΔF/F traces during the tone-onset to puff-onset period (*Figure 3—figure supplement 1A*). The mean peak amplitude for cells from trace conditioned mice was significantly higher than for cells from pseudo-conditioned mice or for spontaneous activity data (trace = 0.022 ± 0.007,

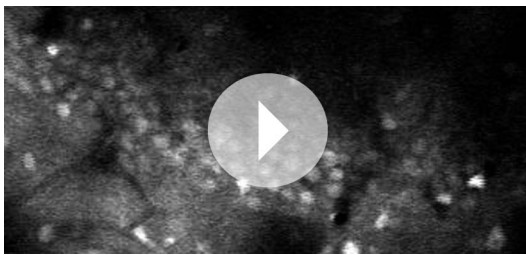

**Video 2**. Calcium responses. Video showing a time-series of images of a single field of view of area CA1 cells. Brighter colors in the gray-scale indicate higher fluorescence intensities. Flashes visible are calcium responses.

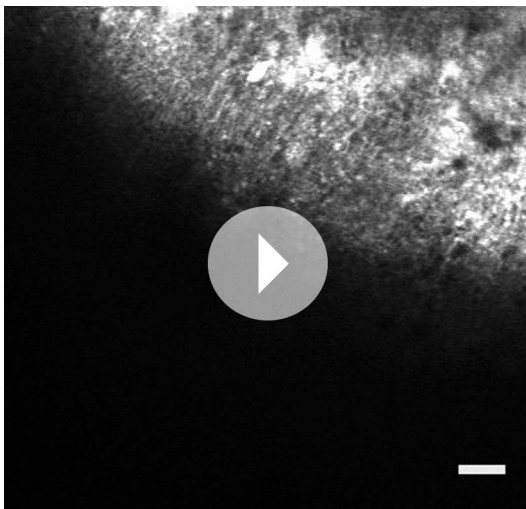

**Video 3**. z-stack of dye-loaded hippocampal tissue. Video showing optical sections through a typical, dye-loaded imaging preparation of the dorsal hippocampus. Optical sections were acquired beginning at the dorsal hippocampal surface and moving ventrally in steps of 2 μm/frame. The scale bar represents 50 μm. The densely-packed cell-bodies of the curved, *stratum pyramidale* cell-body layer appear near the 7 s time-point in the video. The frames of this video show some motion—they were not motion corrected as each frame was taken at a different depth and thus, differed from the others.

pseudo = 0.017 ± 0.009, spontaneous = 0.015 ± 0.008; mean ± SEM; one way Analysis of Variance (ANOVA), followed by Tukey Kramer honest significant difference (h.s.d.) p<0.01). The difference seen was a small one, but this measure does not control for differences in cell response sizes or average activity. Hence, to more rigorously characterize activity timing reliability, we computed a reliability score for each cell.

The reliability score was computed by comparing each cell's time-tuning to that of control data obtained by artificially disrupting any trial to-trial timing relationships in calcium activity. We achieved this by pseudo-randomly shifting single-trial ΔF/F traces for each cell in time (*Figure 3B*). For every cell, a reliability score was then computed as the ratio of the peak area of averaged ΔF/F traces to the peak area of averaged, random time-offset traces (*Figure 3C*). Hence, the reliability score refers to the size of a time-aligned peak, expressed in multiples of that seen after random shifting. To aid in the intuitive understanding of this score, we report here the reliability score of the example neuron whose traces have been shown in *Figure 3A*, which is clearly firing reliably at a particular time (RS = 2.5). That is, the area under the peak of the averaged curve for this cell is more than double that obtained after giving random offsets to individual trials.

Reliability scores for area CA1 cells in mice that learned the task (n = 542 cells from 6 mice), were significantly higher than reliability scores for neurons from pseudo-conditioned mice (n = 498 cells from 6 mice), non-learners (n = 612 cells from 8 mice) and for spontaneous activity data from the trace learners (trace = 3.12 ± 0.20, pseudo = 1.35 ± 0.08, spontaneous = 0.84 ± 0.07, non- learners = 1.72 ± 0.13; mean ± SEM; ANOVA followed by h.s.d., p<0.01; *Figure 3D*). Since the peak timing reliability scores for cells from trace learners were significantly higher than those for controls, we concluded that only neurons from trace conditioned, learner mice had reliably time-tuned peaks in the ΔF/F trace. On plotting the distribution of all reliability scores obtained for cells from trace learners, we found that 44% of

neurons had reliability scores close to 1, that is they displayed no time-locked activity (*Figure 3— figure supplement 1B*). Hence, for all subsequent analysis involving time-tuned activity, 50% of the neurons with the lowest reliability scores were discarded so that we could focus on the cells that did have significant time-tuning. The same treatment (calculation of reliability scores and then discarding the poorest 50%) was also given to control data from the spontaneous activity period, from pseudo-conditioned mice, or from non-learner mice.

Having seen that reliable time-tuning emerges only in trace conditioned mice we next examined the emergence of time-tuning during training. We quantified how the reliability in peak-timing changed as the training session progressed. The same reliability score (*Figure 3A–C*) was calculated for the 50% most reliable neurons in five-trial blocks ('Materials and methods: Data analysis'). We found that the

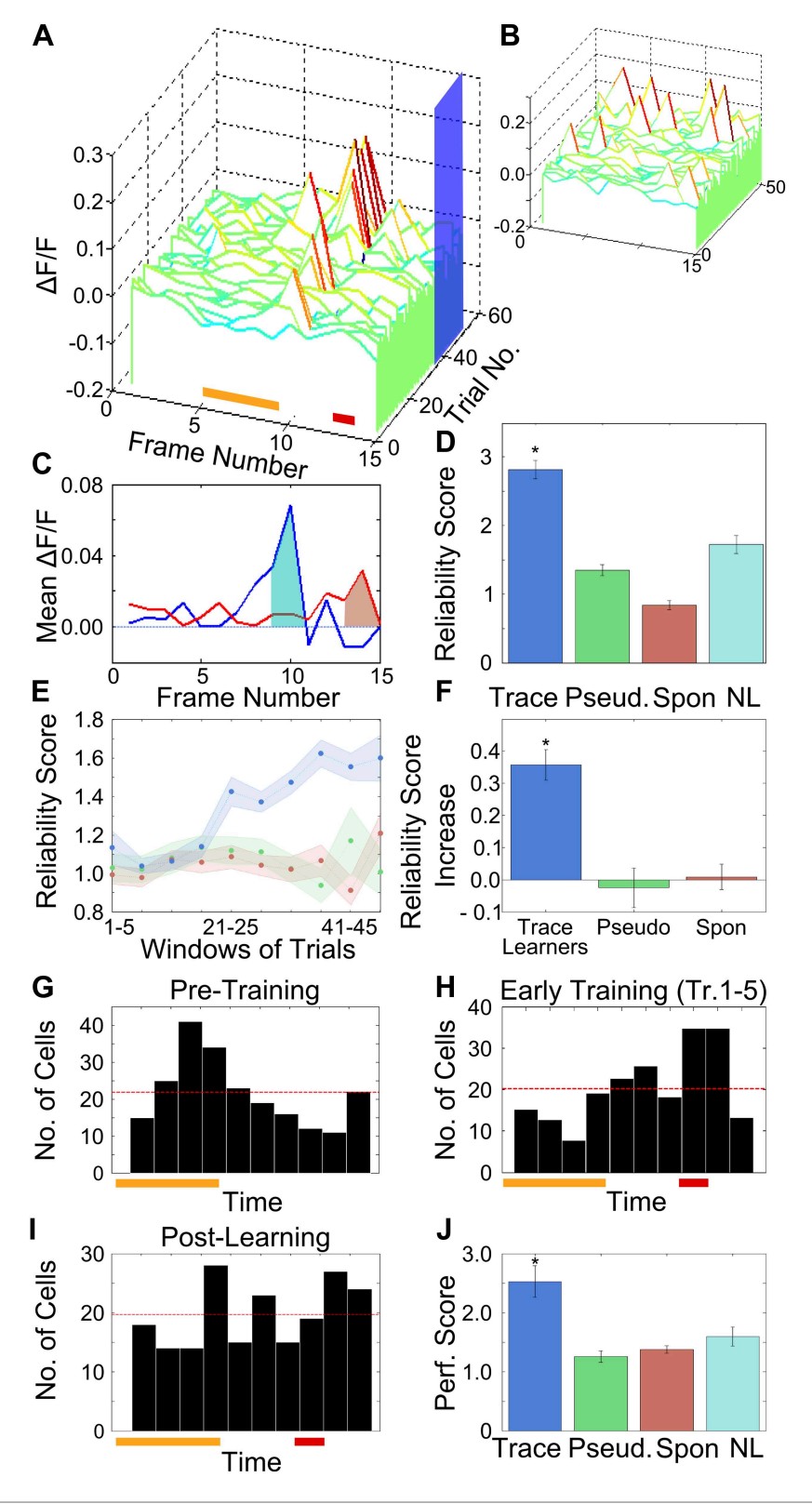

**Figure 3**. After training, area CA1 cells show reliably timed, sequential calcium responses. (**A**) Calcium response traces for an example neuron, aligned to the time of stimulus delivery (tone and puff indicated by yellow and red bars at bottom respectively). Warmer colors in the traces indicate higher ΔF/F values. The blue rectangle on the

*Figure 3. Continued on next page*

*Figure 3. Continued*

'trial number' axis indicates the trial averaging window comprising all trials after the learning trial. (**B**) The same data as in **A**, except with each trial's ΔF/F trace given a random time offset. (**C**) Averaged calcium response curves obtained from aligned, as well as random time-offset traces. The averaged curve from time-aligned traces (blue curve) has a prominent peak, which is absent in the random time offsets case (red curve). This indicates that the neuron fires reliably at a fixed time relative to stimulus delivery. The area under the shaded region (peak ± 1 frame) was used to calculate the reliability score. (**D**) Area CA1 cells from trace learners show significantly higher activity-timing reliability scores. Average reliability scores for all neurons in the entire dataset for learners of the trace-conditioning task (blue), pseudo-conditioned mice (green) spontaneous activity data (red) and data from non-learners (cyan; * indicates p<0.01). (**E**) Change in reliability score with learning for neurons from trace conditioned (blue), pseudo-conditioned (green) and spontaneous activity data (red) respectively. Reliability of firing at the final peak response time (PT) gradually increased over the training session. Reliability scores were computed in five-trial bins. The shaded regions represent SEM. (**F**) Change in reliability scores between early and late blocks of training trials. The increase over the training session for trace-learners was significantly higher than for spontaneous activity data or for pseudo-conditioned mice (* indicates p<0.01). This increase was calculated by subtracting reliability scores of the first half of the session from those of the second half. (**G** and **H**) Distributions of single-cell peak response times (PT) at different stages of learning: pre-training (**G**) and early in training (**H**). The distribution shifts from one showing distinct peaks at the time of tone delivery in the pre-training stage (**G**), to one showing a peak at the time of the air-puff during early training (**H**). Only the PTs of cells with high reliability scores were included in these plots. The red and yellow bars at the bottom indicate times of delivery of tone and air puff respectively. (**I**) Distribution of single cell peak response times (PT) after the learning trial in trace conditioned mice. For trials after the learning trial, PT is uniformly distributed across all times in the trace interval between tone and puff. Only the PTs of cells with significant reliability scores were included in this plot. (**J**) Average, time-decoder performance score, pooled across all trace conditioned mice (blue), pseudo-conditioned mice (green), spontaneous background activity (red) and non-learners (cyan). Dotted line indicates chance level scores, error bars indicate SEM (* indicates p<0.01). *Figure 3—figure supplement 1*, presents further characterization of the reliability score increase. A schematic explaining the time-decoder algorithm is depicted in *Figure 3—figure supplement 2*.

The following figure supplements are available for figure 3:

**Figure supplement 1**. Activity peak times change during learning.

**Figure supplement 2**. Time-decoder algorithm.

---

reliability scores for neurons from trace learners increased over the training session (*Figure 3E*) and that this increase was significantly greater than that seen in pseudo-conditioned mice or in spontaneous activity data from trace conditioned mice (*Figure 3F*, ANOVA followed by Tukey–Kramer h.s.d., p<0.01). Interestingly, the first trial bin where the increase in reliability scores is seen is for trials 21–25. This coincides well with the mean behavioral learning trial (24 ± 1 for imaged mice, 26 ± 2 for all mice mean ± SEM).

The peak reliability score seen in *Figure 3E* is lower than that seen in *Figure 3D*. This is because the number of trials used to calculate the score in each case is different (bin-widths of 5 and 24 respectively). The size of the peaks in the averaged trace from randomly time-shuffled trials depends on the number of trials averaged together since averaging time-shuffled traces would cause isolated peaks to collapse in proportion to the number of trials averaged. On the other hand, peaks aligned in time across trials would reinforce each other upon averaging. We computed reliability scores for different bin sizes and found that mean reliability score increases with the number of trials used to compute it (*Figure 3—figure supplement 1C*). This explains the difference in peak reliability scores seen in *Figure 3D,E*.

We concluded that time-selective area CA1 cell activity emerged progressively, as the mouse learned an association between two stimuli separated in time. Furthermore, the time-course of their emergence matched the average time taken by mice to learn the association.

We next considered two possible phenomena that might account for the improvement of reliability scores with learning. First, a cell may have a fixed activity peak timing from the beginning of the session, but not fire on every trial. In such a case, if the cell gradually started firing at the same, fixed peak-timing on a greater fraction of trials, its reliability score would increase. The cell in *Figure 3A* is such a cell. Alternatively, the timing of peak activity of the cell might change (e.g., the cells in *Figure 2C*).

Such a cell too, would show an increase in reliability score. We therefore computed scores that distinguished between these phenomena, to establish that the predominant effect of learning involves changes in cell activity peak timings, and not their response probabilities.

First, we assessed the extent to which increases in trial to trial response probability affected reliability scores, ignoring changes in peak timing. We calculated local reliability scores as training progressed (*Figure 3—figure supplement 1D*). Here, the timings at which reliability scores were calculated were re-estimated for each five-trial bin. Hence, if peak-timing were to shift during training, it would not affect the reliability score. On the other hand, if cells began to show activity on a larger proportion of trials, the score would increase. On average, no training-related increases in local reliability score were detectable. This indicates that most cells do not increase their response probability with training.

We next quantified how local activity peak timings for each neuron differed from the peak timings found after learning. Half the trials after the learning trials were used to estimate post-learning activity peak timings for all cells. Then, we computed local peak timings for 10-trial bins through the entire session and computed the absolute difference between these local peak timings and the post-learning peak timings. These differences served to estimate how far, on average, cell peak timings were from their values after learning. We observed a gradual decrease in absolute peak timing difference as training progressed (*Figure 3—figure supplement 1E*) for cells from learners. Such a decrease was not seen for peak timing differences computed for cells from pseudo-conditioned mice, spontaneous activity data or non-learners. We also calculated the same peak timing differences, binning all trials before and after learning. The difference in peak timings from the post-learning peak timings was high prior to learning, and reduced to the order of a single frame-time post learning (*Figure 3—figure supplement 1F*) for trace learners. For mice that learned the association, the absolute difference in peak timings prior to, and post learning, were significantly different (ANOVA followed by Tukey Kramer h.s.d, p<0.01). No such decrease in peak timing difference was seen for any of the control cases. Hence, we concluded that reliability scores increase as learning progresses and that this increase is mainly due to changes in the peak-timings of cell activity during learning.

To characterize the changes in peak timings of cell populations due to learning, we next examined the distributions of the PTs of all neurons pooled across trace conditioned mice, at various stages of learning. The distribution was clearly non-uniform both before ($\chi^2$ goodness of fit test with uniform expected distribution; $\chi^2 = 69.14$, $p<10^{-4}$, *Figure 3G*) as well as at an early stage of trace conditioning ($\chi^2 = 61.92$, $p<10^{-4}$, *Figure 3H*). However, for the group of trials after the learning trial, the distribution was much closer to a uniform distribution ($\chi^2 = 20.53$, $p=0.015$, *Figure 3I*). Thus, the highly non-uniform PT distributions prior to and early in training became more uniform, effectively tiling the entire period of interest as a result of trace conditioning.

If neuronal activity is indeed sequential and time-locked, it should be possible to use activity data to decode time since tone onset. Thus, as an independent test of time-locking of activity, we formulated a simple time-decoding algorithm based on template matching (*Fenton and Muller, 1998*; *Zhang et al., 1998*; 'Materials and methods: Data analysis'; *Figure 3—figure supplement 2*). The decoder uses single-frame, ΔF/F fluorescence values from cell-populations on single trials to decode time elapsed since tone presentation. The time-decoder prediction accuracy scores (normalized to decoder accuracy for control, time-shuffled data) for data from trace learners were significantly higher than those for data from pseudo-conditioned mice, spontaneous activity data from trace learners and data from non-learners (trace = 2.28 ± 0.27, pseudo = 1.26 ± 0.10, spontaneous = 1.37 ± 0.06, non-learners = 1.59 ± 0.16, mean ± SEM; ANOVA, followed by Tukey Kramer h.s.d., p<0.01; *Figure 3J*). As with the activity timing reliability score, the time decoder prediction score is also expressed in multiples of the score achieved using temporally randomized, control data. Hence, a decoder score of 2 would indicate prediction capability twice as good as that obtained with randomized data. To aid an intuitive understanding of the score, we also calculated the percentage of decoder frame-number predictions that were within one frame of the correct frame. For activity data from trace learners, the percentage was 70.3%, for pseudo-conditioned mice it was 32.4%, for spontaneous activity data it was 47.6%, and for non-learners it was 53.1%. The mean correct prediction percentage by random chance was 28.8%. The mean performance scores reported in *Figure 3J* closely match the values obtained by taking the ratio of these prediction percentages to the chance prediction percentage.

We also checked if neurons sharing time-tuning were spatially clustered within the hippocampus. The mean pair-wise distance between neurons sharing time-tuning was not significantly different from the mean distance between random pairs of neurons, that is neurons sharing time-tuning were not

spatially clustered together (mean ± SEM distance for same time-tuning 71.27 ± 1.53 μm, random 70.89 ± 0.62 μm; *Figure 3—figure supplement 1G,H*; two-sample *t* test p=0.73).

Together, these data suggested that prior to training, area CA1 cells responded in a manner that was stimulus locked. These cells began to show reliably timed, sequential activity when mice were learning a task that required the maintenance of a sensory representation through a trace time-interval. Training caused a shift in the distribution of these activity peak times towards more uniform coverage of the interval of interest, spanning tone and puff stimulus durations as well as the intervening trace period.

## Noise-correlations increase transiently during learning, especially between similarly time-tuned neurons

Trial by trial correlations in spontaneous activity (noise correlations or NC) have been used previously as a measure of functional connectivity and have been thought to indicate the presence of direct (monosynaptic) or indirect, polysynaptic connections between correlated neurons (*Ts'o et al., 1986*; *Bair et al., 2001*; *Fujisawa et al., 2008*). Noise correlations, as measured by slower calcium recordings, are also thought to be indicative of shared presynaptic input, and therefore, changes in correlations are thought to imply changes in common input (*Komiyama et al., 2010*). We calculated correlation coefficients between the spontaneous activity traces of pairs of neurons (more than 4 s away from any stimuli), (*Figure 4A*, dashed, blue curve). Interestingly, for trace conditioned learners, mean NC increased during the training session, with a peak at trial bin 16-20 (19 ± 2, mean ± SEM). This peak precedes the peak in time-tuning (trial bin 21–25, *Figure 3E*) and the averaged learning trial (24 ± 1 for imaged mice, trial 26 ± 2, for all mice; mean ± SEM). This relationship was also observed at the level of individual mice (*Figure 4—figure supplement 1A*, mean learning trial = 24 ± 1, trial of peak reliability score = 25 ± 2, peak NC trial = 19 ± 2 mean ± SEM; ANOVA p=0.03).

Furthermore, the NC calculated for cell-pairs where both cells had the same activity peak timing was significantly higher than for random cell-pairs (same time-tuning = 0.22 ± 0.01, random pairs = 0.14 ± 0.01 (mean ± SEM), ANOVA followed by Tukey–Kramer h.s.d., p<0.01, n = 56 peak-timing points with an average of 25 cell-pairs in each group; *Figure 4B*). Towards the end of the session, NC declined, although, even at this stage, NC for neurons sharing the same peak times remained significantly higher than for randomly chosen cell pairs (same PT = 0.16 ± 0.01, random pairs = 0.10 ± 0.01; mean ± SEM; ANOVA followed by Tukey–Kramer h.s.d., p<0.01; *Figure 4B*). Cells from pseudo-conditioned mice showed no such increase in NC (same PT = 0.13 ± 0.02, random pairs = 0.12 ± 0.01; mean ± SEM; ANOVA followed by Tukey–Kramer h.s.d., p>0.01; n = 48 time-tuning points with an average of 25 cell-pairs in each; *Figure 4B*).

We also checked if the spontaneous activity correlations were higher for cells with high activity peak timing reliability scores. Neurons with high reliability scores had significantly higher noise correlations than neurons with low reliability scores (*Figure 4—figure supplement 1B*; high RS 0.24 ± 0.03, low RS 0.14 ± 0.02; mean ± SEM; * indicates p<0.01, two sample *t* test).

Finally, we checked if activity sequences seen during the stimulus period were re-capitulated during spontaneous activity. We gave the spontaneous period activity trace of each neuron a time-offset equal to the relative timing of its activity peak timing within the stimulus period sequence. Then, we calculated the correlation coefficients between all such peak-time offset traces. If the same sequence indeed recurred during spontaneous activity, these correlations would be higher. As a control, we gave these spontaneous activity traces randomly chosen time-offsets from the set of peak-timings observed. Correlations for peak-timing offset traces were not significantly different from those for randomly offset traces (*Figure 4—figure supplement 1C*; peak-time offset 0.048 ± 0.0012, random offset 0.047 ± 0.0009, means ± SE; two sample *t* test, p=0.71). However, it must be pointed out that if sequence replay events are rare in comparison to spontaneous activity of area CA1 cells, then the correlation analysis described above, coupled with the limited size of our dataset (to minimize calcium indicator dye photo-bleaching) might not have the statistical power to detect them. Furthermore, spontaneous, sequential replay activity could still be occurring, but at an accelerated time-scale, as has been observed before (*Lee and Wilson, 2002*). Such fast replay events would not be resolvable, given our temporal resolution capability.

Surprisingly, spontaneous activity correlations for trace learners were observed to drop towards pre-training levels in the second half of the session (*Figure 4A,B*). This has been observed previously in the context of rats repeatedly exploring a novel track (*Cheng and Frank, 2008*). Here, the reduction

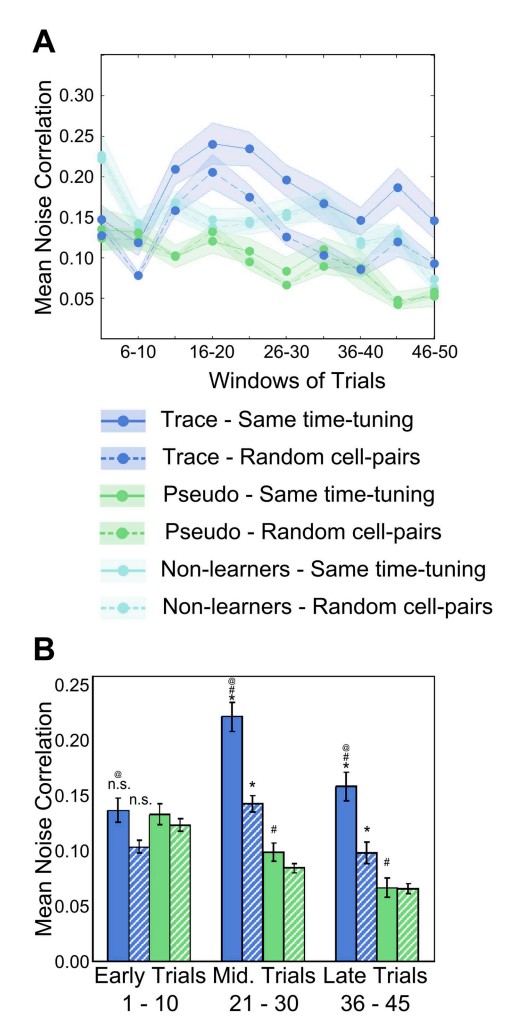

**Figure 4**. Area CA1 cell noise-correlations increase transiently during training. (**A**) Average neuron-pair noise correlations plotted as a function of training trials for task learner mice (blue curves), pseudo-conditioned mice (green curves) and non-learners (cyan curves). Solid lines represent average noise correlations between neurons that share similar time tuning (same PT), whereas dashed lines indicate average noise correlations between random neuron pairs. Pair-wise noise correlations have been determined from spontaneous activity traces over five trial windows. Shaded regions indicate SEM. (**B**) Summary statistics comparing average noise correlations across early (trials 1–10), middle (trials 21–30) and late (trials 36–45) stages of training, between task learner mice (blue bars) and pseudo-conditioned mice (green bars). Solid bars represent average noise correlations between neurons that share similar time tuning (same PT) whereas hatched bars represent average noise correlations between random neuron pairs. Error bars represent SEM. (* indicates within condition [same time-tuning v/s random cell-pairs], # indicates across conditions [trace

*Figure 4. Continued on next page*

in spontaneous activity correlations was ascribed to a reduction in the novelty of the particular congruence of stimuli related to the track being explored. Hence, the reduction in noise correlations we observe could also be a post-learning phenomenon, related to a reduction in the novelty of the specific congruence of the paired CS and US stimuli. To support this interpretation, we ruled out possible technical contributors to the reduction in correlations. First, the reduction in correlations were not due to a gradual decline in the quality of optical activity recordings, since we found that recordings remained stable through the training session, as discussed in 'Results: Cells in hippocampal area CA1 imaged from awake, behaving mice exhibit temporal tuning'; *Figure 2— figure supplement 1D,E*. Second, this effect was not due to mice paying less attention to the task or fatiguing towards the end of the session. We tested this by measuring the sizes and peak latencies of the significant blinks before and after trial 25, where the decline in spontaneous correlations was seen to begin. Mean significant blink sizes before and after trial 25 were not significantly different (mean normalized early blink size = 1.03 ± 0.06, late = 0.95 ± 0.03, mean ± SEM; two sample $t$ test p=0.62; *Figure 4—figure supplement 1D*). Additionally, mean significant blink peak latency from the time of tone onset did not change from the first to the second half of the session (early latency = 565.3 ± 68.5 ms, late latency = 538.7 ± 92.3 ms; mean ± SEM; two sample $t$ test p=0.80; *Figure 4—figure supplement 1E*). Hence, we concluded that blink responses remained stable across the two halves of the session, and that a lack of sustained attention was not likely to be the cause of the reduction in noise correlations.

To summarize, spontaneous activity was seen to become more correlated as a result of trace conditioning, following which time-tuning peaked and mice learned the temporal association. Furthermore, neurons that eventually shared the same time-tuning, showed the greatest increases in NC.

## Correlated cell-groups are organized into spatial clusters that are re-organized by training

We next checked if area CA1 cells could be separated into groups sharing high within-group NC and low across-group NC. We used the previously described 'meta k-means' clustering algorithm to identify groups of neurons that showed highly correlated spontaneous activity (*Figure 5A–D*; *Ozden et al., 2008*; *Dombeck et al., 2009*). We observed that neurons clustered into

*Figure 4. Continued*

learners v/s pseudo-conditioned] and @ indicates comparisons across stages of learning [early v/s middle v/s late], p<0.01; n.s. indicates not significant). *Figure 4—figure supplement 1A* depicts the point in the session at which behavioral CR rates, CA1 cell timing reliability and spontaneous activity correlations reach their peaks, on an individual mouse basis. It also presents further characterization of the changes in NC during learning.

The following figure supplements are available for figure 4:

**Figure supplement 1**. Peaks of CR rate curve, CA1 activity timing reliability and NC; characterization of NC changes during learning.

distinct groups. As expected, within-cluster NC was higher than across cluster NC (*Figure 5B,D*, two-sample $t$ test, p<10$^{-4}$). Interestingly, in contrast to the groups of neurons sharing the same activity peak-timings (*Figure 3—figure supplement 1G,H*), neurons belonging to the same correlation cluster were spatially clustered within the imaged fields of view (examples of peak-timing groups and correlation clusters are shown in *Figure 5E*). The mean distance between neuron centroids for pairs of neurons belonging to the same cluster was significantly lower than for pairs of neurons taken from different clusters (intra-cluster = 72.10 ± 1.00 μm, n = 1694 pairs, inter-cluster = 106.79 ± 0.90 μm, n = 3418 pairs, mean ± SEM; two-sample $t$ test p<10$^{-4}$, *Figure 5F*). Furthermore, NC and cell–cell distance were also found to be significantly, inversely correlated (correlation coefficient = −0.178, p<10$^{-4}$, n = 24,522 cell pairs; *Figure 5—figure supplement 1A*). This indicates that neuronal NC are indeed spatially organized.

To examine how these cell-groups or clusters changed over the session, we calculated a similarity score for clusters identified from different sections of the training session ('Materials and methods: Data analysis'). We considered spontaneous activity data from three segments of the session comprising early, middle and late blocks of trials. We then measured the similarity between pairs of matched clusters obtained using data from different trial blocks. The similarity score measured the mean fraction of neurons belonging to clusters matched across these blocks, normalized to the mean overlap between random cell-clusters of the same sizes. For example, a high similarity score for a cluster (>> 1) would indicate a similarity much greater than chance, implying that this cluster is well preserved from one trial block to the next. Interestingly, we found highly reliable groupings (i.e., high similarity scores) in the pre-training session datasets, indicating stable patterns of input from upstream neuronal populations (example in *Figure 6A*) prior to training. In contrast, cell-groupings changed drastically during the trace conditioning session (*Figure 6A,C*), and were significantly less stable than those in the pre-training datasets. For pseudo-conditioned mice, cell groupings were reliable throughout the training session datasets (ANOVA followed by Tukey Kramer h.s.d., p<0.01; *Figure 6B,C*). Thus, spatially organized, correlated cell-clusters exist prior to and during learning. These correlated cell-groups remain stable in the absence of training, but undergo continuing re-organization during the process of learning the trace conditioning task.

Are groups of time-tuned neurons organized into these correlation clusters? We calculated a cluster similarity score for neurons grouped by correlated activity and neurons grouped by time-tuning, during the tone to puff period. We found that this similarity was close to chance (mean overlap score = 2.08, SD = 0.6, in multiples of similarity expected by chance; 'Materials and methods: Data analysis').

## Discussion

We showed that, during trace eyeblink conditioning, groups of area CA1 neurons progressively began to fire in sequences that spanned the interval from tone onset to 200 ms after the puff (*figures 2D, 3D,E*). This phenomenon became prominent at the same trial in the session at which mice learned the association (*Figure 1G*). The emergence of activity sequences and learning were closely preceded by a transient increase in noise-correlations, and was accompanied by modifications to spatially organized clustering of correlated neurons (*Figure 4A,B, 5F, 6A–C*). As has been seen before in the context of place cells, we found that neurons related to sequential activity were not spatially organized within the hippocampus (*Dombeck et al., 2010*; *Figure 3—figure supplement 1F,G*).

### Stimulus representation by activity sequences–the 'trace' in trace eyeblink conditioning

The objective of this study was to shed light on the role played by the hippocampus in the association of stimuli separated in time, specifically during trace eyeblink conditioning. Previous work had identified

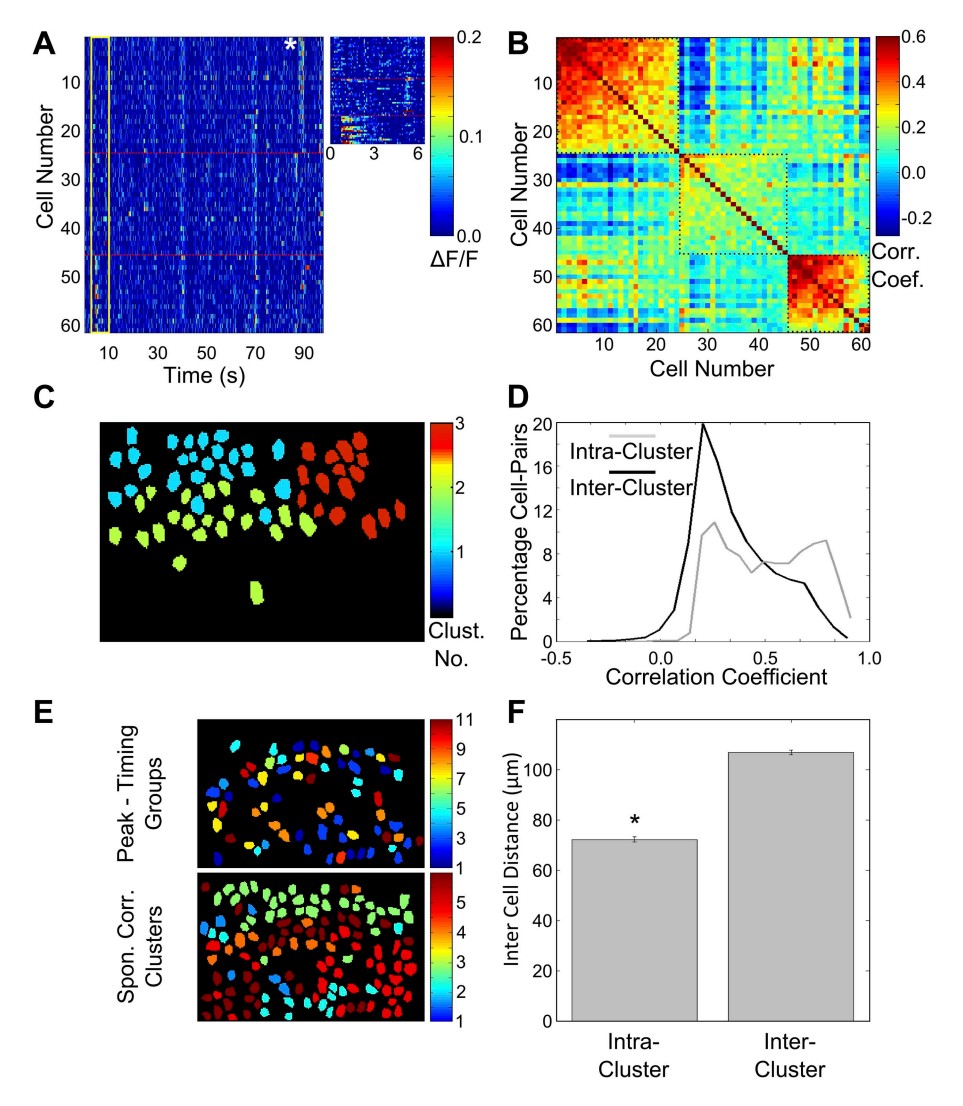

**Figure 5**. Correlated neurons are spatially clustered. (**A**) Neurons within clusters show within-group, correlated activity. Spontaneous, ΔF/F calcium activity traces for neurons from an example dataset. Neurons are sorted as per clusters identified by meta k-means clustering. Cluster boundaries are indicated by red, dotted-lines. The color scale represents ΔF/F amplitude. A short, 6 s stretch of the traces outlined by the yellow box have been re-plotted on the right for clearer visibility. The white asterisk indicates a bout of correlated activity in a cluster. (**B**) Pair-wise noise correlation matrix for neurons sorted as per cluster identity, for the dataset in (**A**). Color scale represents the pairwise correlation coefficient during spontaneous activity (noise correlation). Boxes indicate within-cluster spontaneous activity correlations. (**C**) Masks of neuron ROIs for the dataset shown in **A** and **B**, color coded as per cluster identity. (**D**) Comparison of distributions of within cluster (gray) and across cluster (black) correlation-coefficients. The distributions are significantly different ($p < 10^{-4}$). (**E**) The same field of view, with neuron ROIs color-coded by timing of their activity peak in post-training trials (top) and by correlation cluster number (bottom). The number of cells in the top panel is smaller as only 50% of the cells with the highest reliability scores were found to be time-tuned. (**F**) Correlated cell-clusters are spatially organized. Inter-cell distances or the distances between the centroids of pairs of neurons, when both belong to the same noise-correlation cluster (Intra-Cluster) or when they belong to different clusters (Inter-Cluster). Error bars indicate SEM. (* indicates $p < 10^{-4}$). ***Figure 5—figure supplement 1*** shows that the spontaneous activity correlations between pairs of cells fall off with increasing distance between the cells being considered.

The following figure supplements are available for figure 5:

**Figure supplement 1**. Spontaneous activity correlations fall with increasing inter-cell distance.

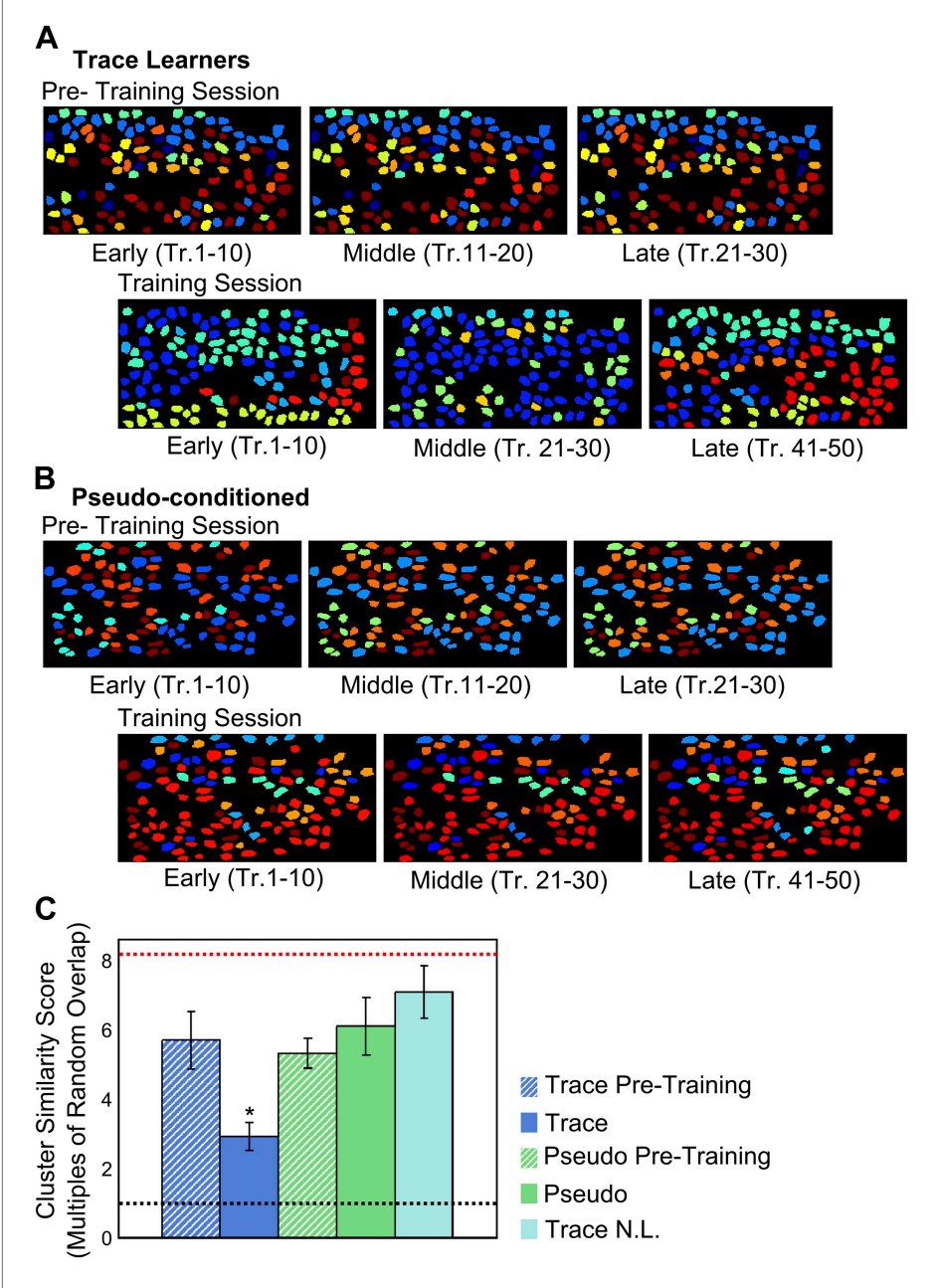

**Figure 6**. Correlated neuron-groups are spatially re-distributed during trace conditioning. (**A** and **B**) Sample trace-conditioned (**A**) and pseudo-conditioned (**B**) mouse neuron-clusters, identified in early (left), middle (center) and late (right) trials, for pre-training (top) as well as training (bottom) sessions. Neurons have been color-coded as per the noise-correlation cluster they belong to, and cluster numbers have been sorted for maximum cluster member overlap across trial-groups. (**C**) Spatial organization of correlated cell-clusters changes during learning. Summarized statistical testing of stability of clusters over the training session. A cluster similarity score (SS) was computed to measure the similarity between clusters from different parts of the session. Mean similarity scores for clusters of neurons from trace conditioned mice are shown in blue, pseudo-conditioned mice in green, and non-learners in cyan. Pre-training session scores are displayed as hatched bars and training session scores, solid bars. Error-bars denote SEM. The black, dotted line denotes chance level overlap score, and the red, dotted line indicates mean overlap score with perfectly overlapping clusters (* indicates p<0.01).

neurons that were primarily air-puff responsive early in training, but became progressively more tone responsive as training proceeded. This process was thought to ascribe US valence to the CS (*McEchron and Disterhoft, 1999*). The progressive changes in neuronal activity timing we observed are consistent with this finding, as a small subset of neurons did indeed shift from being tone to puff responsive (*Figure 2E,F*, *Figure 3G–I*). However, the predominant learning-related change we observed was the emergence of sequentially timed activity of groups of neurons (*Figure 3D,E,J*, *Figure 3—figure supplement 1D–F*). Here, we note that calcium response recordings typically have a time resolution of about 100 ms, a time window within which entire sequences of neuronal activation have been known to occur (*Foster and Wilson, 2006*). Hence, we interpret our data to be indicative of sequential activity with respect to time-windows comparable to the average frame time (74 ms) of our recordings. Even at our lower time resolution, the total space of possible sequences is very large. Our study thus indicates that a very small fraction of all possible sequences is important for successful trace conditioning. Grouped by activity peaks within ~70 ms time-windows, neurons might show finer-scale sequences or even fluctuations that would not be visible in our recordings. However, at the time-scale of a few hundred milliseconds relevant to trace conditioning, area CA1 cells show progressively emerging activity sequences after network re-organization during learning.

Another suggested model for stimulus trace representation involves sustained neuronal activity (*Solomon et al., 1986*). We saw no evidence to support this model. Our observation of the progressive formation of cell activation sequences (*Figure 2C,D, 3E,J*) significantly extends our understanding by revealing how the representation of the tone stimulus is maintained over the tone-puff interval in the trace conditioning task.

Time-tuned firing has previously been interpreted to imply that neurons encode time (*MacDonald et al., 2011*) or that they encode a triggering stimulus in the form of a well-timed sequence, effectively bridging two temporally separated stimuli (*Itskov et al., 2011*; *MacDonald et al., 2013*). Stimulus representation by sequential activity would also allow for the association of two temporally separated stimuli, by maintaining a representation of the first stimulus until the arrival of the second one. This would then allow Hebbian plasticity to strengthen synapses linking the neurons associated with the non-stationary, sequential stimulus representation. This stimulus maintenance is necessary since Hebbian plasticity cannot occur with stimuli separated by over ~100 ms (*Levy and Steward, 1983*).

Simulation studies of the hippocampal CA3 network, have shown that stimuli separated in time can be 'bridged' (*Wallenstein et al., 1998*). Simple recurrent network models of the hippocampal CA3 region have also been trained on a close proxy of the trace conditioning paradigm (*Howe and Levy, 2007*; *Levy et al., 2005*). Notably, in these studies, the tone stimulus triggers a well-timed relay of CA3 cell activation that culminates in the activation of US representing neurons. Our experimental observations are in agreement with this model, if we assume that qualitatively, grouping in area CA1 cell activity reflects grouping of upstream CA3 activity (*Brivanlou et al., 2004*). We suggest that the *progressive* emergence of sequentially timed activity we observe in our study is a signature of such network changes.

Sequential activity has been observed before, in subjects that have already learnt a temporal memory task (*Lee and Wilson, 2002*; *Pastalkova et al., 2008*; *MacDonald et al., 2011*). While sequentially-timed activity has also been observed in un-trained mice (*Luczak et al., 2009*; *Dragoi and Tonegawa, 2011*), these sequences of activation were very short, only 100–200 ms long. Post training, we observed sequential activation over an 800 ms period, which is a task relevant time-scale (*Figure 2D*). Importantly, to our knowledge, our study is the first to monitor the emergence of these sequences over the entire process of learning a temporal memory task (schematic in *Figure 7A*). As we discuss below, this has allowed us to identify transient, learning-related network changes in spontaneous activity correlations that could underlie the emergence of the longer lasting sequential activity.

As discussed earlier ('Results: Mice learn a trace eyeblink conditioning task within a single session'), area CA1 activity data pooled from non-learners is probably confounded by contrary effects due to the uncertain state of learning of each mouse. However, for the sake of completeness, we have included data from non-learners in some key figures. These panels show that the data from non-learners is consistent with our interpretations and that trace learners are significantly different from non-learners as well as from pseudo-conditioned mice, in all cases.

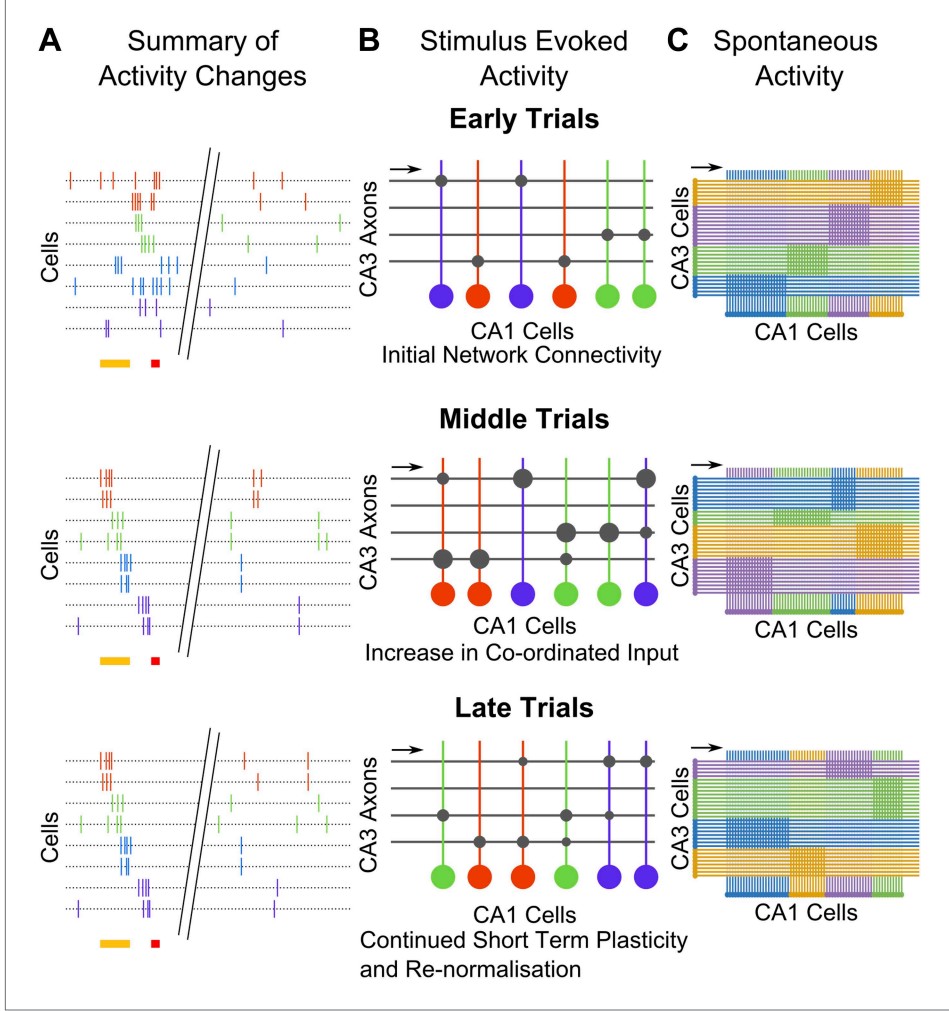

**Figure 7**. Summary and model schematics. (**A**) Schematic summarizing findings: early in the session (top), neuron activity peaks are tuned to stimuli. Spontaneous activity is largely un-correlated. By the middle of the session (center), time-tuned activity during the stimulus period begins to emerge, after spontaneous activity correlations have risen. At the end of the session (bottom), stimulus–period activity is time-tuned, but spontaneous correlations have fallen back towards baseline levels. (**B**) Cartoon model of possible network changes occurring at the synaptic level: early in training (top), CA3 to CA1 synapses (gray circles) are at baseline strengths. As training progresses, specific synapses are strengthened (larger gray circles), towards the middle of the session (middle). At this point, neurons that share time tuning also begin to display increased noise-correlations as they receive more common input. Late in the training session (bottom), after repeated re-organization of the network, synapses undergo homeostatic normalization (gray circles reduced in size), thus causing total common input and spontaneous activity correlations to drop. (**C**) Schematic diagram depicting network changes at the level of groups of cells. CA1 cell clusters (horizontal row at bottom of each panel; clusters are color-coded) have high correlations in spontaneous activity, receiving shared input (direction indicated with black arrows) from groups of CA3 cells that are also correlated in their spontaneous activity. These correlation-groups of cells are spatially clustered. As learning progresses, the clusters of correlated CA3 cells change, resulting in changes in the groups of correlated CA1 cells. These new groups are also spatially organized, but differ significantly from pre-training groups.

## Network reorganization during learning is reflected in changes in spatially organized noise correlations

Correlations between neurons during periods of spontaneous activity or noise correlations (NC) have been thought to be indicative either of shared inputs, or of direct or indirect connectivity (*Ts'o et al., 1986*; *Bair et al., 2001*; *Fujisawa et al., 2008*). While high temporal resolution measurements of neuronal activity are required to infer direct, synaptic connectivity, we, in our study, infer changes in NC to

be indicative of changes in clustering of inputs to CA1 cells from upstream CA3 cells (*Komiyama et al., 2010*). This view is consistent with earlier findings where the correlations between place cell pairs were seen to increase as rats explored a novel track (*Cheng and Frank, 2008*). Importantly, this increase in correlations was dependent on the expression of N-methyl-D-aspartate receptors and, therefore, the capacity for synaptic plasticity in area CA3 (*Dragoi and Tonegawa, 2013*). We found that cell-pairs that eventually share the same time-tuning show significantly higher increases in NC than do random cell-pairs, though on average, all cell-pairs show a significant increase in pair-wise correlations (*Figure 4A,B*). We interpret this to be due to increased common input from changing groupings of inter-connected CA3 neurons. However, this increase is temporary as average NC declines near the end of the session, although neurons sharing the same time-tuning still maintain significantly elevated NC. This is consistent with a previous observation that novelty-induced increases in spontaneous correlations gradually reduce with repeated exposure to the novel environment (*Cheng and Frank, 2008*). The transient, global increase in NC may arise from a network-wide increase in synaptic weights during learning, and the subsequent decline may reflect weakening of mean synaptic weights due to synaptic re-normalization (schematic in *Figure 7B*). Consistent with this interpretation, we observed that mean NC peaked just prior to the peaks in behavioral responsiveness and reliability of sequential activity, even on an individual mouse basis (*Figure 4—figure supplement 1A*).

Similar results have also been observed in previous studies. Cortical neurons sharing task-specific tuning were seen to increase their spontaneous correlations as learning progressed on a sensory association task (*Komiyama et al., 2010*). In recordings from rodent CA1 neurons, place cells were seen to show gradually increasing spontaneous correlations as a novel environment was explored (*Cheng and Frank, 2008*; *Dragoi and Tonegawa, 2013*). Furthermore, over extended exposure to the same novel track, these elevated correlations gradually reduced (*Cheng and Frank, 2008*).

Interestingly, when neurons were divided into clusters that shared high NC, these clusters turned out to be spatially organized even prior to training (*Figure 5E,F*). This might indicate that area CA1 cells receive spatially segregated input from bundles of area CA3 cells. This is borne out by the observation of spatially segregated connectivity from CA3 to CA1 cells in hippocampal slices (*Brivanlou et al., 2004*). Importantly, we observed that these spatially segregated clusters changed only during learning, while in pseudo-conditioned mice and in data collected prior to training, NC clusters remained relatively constant (*Figure 6C*). This indicates that during learning, area CA1 cells (and probably also, by extension, CA3 cells) enter a dynamic state, where neurons change from one spatially-organized, correlated grouping to another (schematic in *Figure 7C*). In our view, these fluctuating, spatial groupings reflect network changes that underlie the emergence of spatially un-organized, behavior-related sequentially timed activity.

We propose that when learning the trace eyeblink conditioning task, and perhaps more generally when associating temporally separated stimuli, the hippocampus is recruited as follows:

Through a network-wide, transient increase in specific synaptic strengths within the CA3 and in the CA3 to CA1 network, changing groups of CA3 cells remodel the strengths of their connections to CA1 cells (*Figure 7B,C*). This is reflected in changing clusters of area CA1 cells showing increased noise-correlations. Progressively, these groups stabilize into a sequence of stimulus-activated cell-groups, after which global synaptic re-normalization causes average noise correlations to move back towards baseline levels (*Figure 7C*). Only neurons sharing the same timing in the sequence maintain significantly elevated noise correlations (*Figure 4B*, schematic in *Figure 7B*). Importantly, these spontaneously correlated clusters of cells are also spatially clustered and changes in these clusters underlie the emergence of sequentially activated cell-groups. These sequential groups allow CS-representing neurons to be indirectly connected to US-representing neurons, thereby forming an association between stimuli separated in time.

## Materials and methods

### Mice and surgical procedure

All experimental procedures were approved by the National Centre for Biological Sciences Institutional Animal Ethics Committee (Protocol number USB–19–1/2011), in accordance with the guidelines of the Government of India (animal facility CPCSEA registration number 109/1999/CPCSEA) and equivalent guidelines of the Society for Neuroscience. All recordings and behavioral experiments were carried out on male, 30 to 45 day old C57BL/6 mice. Data acquired from a total of 35 mice is included here. A

total of 18 mice were trace conditioned, and 17 mice pseudo-conditioned. Of the 18 trace conditioned mice, 9 learned the task to criterion and the other 9 failed to learn. Imaging data was acquired for 14 of the trace conditioned mice. Of these, 6 were learners and 8 were non-learners. Of the 17 pseudo-conditioned mice, imaging data was acquired for 6 mice (*Figure 2—figure supplement 1B*).

## Surgical Procedure

Surgery was carried out on a mouse anaesthetized using isoflurane (Abbott, North Chicago, IL, USA) gas anesthesia (3% induction dose and 2% initial maintenance dose (in 95% $O_2$, 5%$CO_2$), gradually reduced over a ~3 hr long procedure). Temperature was maintained using a heating-pad (TR-200, Fine Science Tools, Canada). Dexamethasone 5 mg/kg body weight was injected sub-cutaneously to minimize swelling and bleeding prior to the surgery. The mouse was then head-fixed (Item 51547; Stoelting, Wood Dale, IL, USA) and a custom designed brass head-bar was affixed to the skull using skull-screws (83,438; Micro-Mark, Berkeley Heights, NJ, USA) and dental acrylic. The dorsal hippocampal surface (ipsi-lateral to the side of tone and air-puff delivery) was exposed through a craniotomy (2.0 mm posterior to bregma, 1.5 mm lateral to the mid-line [ipsi-lateral to the side of stimulus delivery]; diameter ~ 1–1.5 mm) after gently aspirating away over-lying cortex to expose the external capsule (*Dombeck et al., 2007*). Exposed brain-matter was constantly irrigated with cortex buffer (CB; in mM: NaCl–125, KCl–5, glucose–10, HEPES–10, CaCl2–2 and MgCl2–2. pH set to 7.35 with NaOH) (*Holtmaat et al., 2005*). A fluorescent indicator of calcium (O6807; Oregon Green BAPTA-1 AM ester, Molecular Probes - Life Technologies, Grand Island, NY, USA) was bolus loaded by pressure injection into the hippocampus, at a depth of ~100 μm. Pressure injections were delivered using glass pipettes with tip impedances of 2–3 Mohms and pressure pulses of 5–10 psi for 3 to 5 min (Picospritzer-III, Parker-Hannifin, Cleveland, OH, USA). Dye solution was prepared by solubilizing 50 μg of dye (one vial) in 5 μl of a detergent (P-3000MP; 20% Pluronic acid, Invitrogen - Life Technologies), and then diluting down to 1 μg/μl by adding 45 μl of CB. The gap left behind by aspirated cortex was then filled by pouring molten, low-melting agarose solution (A9414; 5% in CB, Sigma, St. Louis, MO, USA) into the craniotomy and covering with a 5 mm, circular cover-slip (CS-5R; Warner Instruments, Hamden, CT, USA). The cover-slip was then fixed in place with dental acrylic. A fragment from a freshly shattered Neodymium magnet was affixed to the left, upper-eyelid using cyano-acrylate glue. Dexamethasone (5 (± 2) μg/g body weight) was injected sub-cutaneously to minimize swelling and bleeding at the beginning of the surgery. Fortwin (Pentazoncine, 5 μg/g body weight) was administered sub-cutaneously at the end of the surgery, as an analgesic. The mouse was then released and given 95% $O_2$ to breathe until vigorous toe-pinch responses were seen and was then transferred to a closed cage with food and water for 1.5–2 hr, to recover before imaging began. OGB-1 dye, once loaded into neurons, allows imaging for a limited period before clearance from neurons (*Stosiek et al., 2003*). This limited the viability of the imaging preparation to a few hours, corresponding to a single behavioral session.

## Behavioral training

1.5 to 2 hr after recovery from anesthesia, mice were head-fixed on a custom-designed clamp (gift from the Albeanu Lab, Cold Spring Habor Laboratory, New York, NY, USA) and mounted onto the stage of a custom-built, two-photon microscope. Left eyelid blink responses were recorded using a custom-made magnetometer (*Koekkoek et al., 2002*) positioned close to (1–3 mm away) the neodymium magnet fragment glued to the eyelid and was left un-disturbed throughout the experiment. The magnetometer was constructed using a magneto-resistive sensor (HMC1051Z; Honeywell, Linden, NJ, USA), whose output was further amplified (SR560; Stanford Research Systems) and digitized at 10 KHz using a data acquisition card (PCI-6221; National Instruments, Austin, TX, USA). The tone, conditioned stimulus was delivered using a speaker positioned to the left of the mouse. The stimulus was a pure tone of frequency 9.5 KHz, delivered at an intensity of 80–90 dB at the microscope stage and of 350 ms duration. In a pre-training session, only tones were delivered for 30 trials. Blink responses were acquired for 5 s prior to tone onset and terminated 10 s after tone onset. The inter-trial interval was randomized between 15 and 30 s, making the total duration between tone-presentations 30 to 45 s. Thereafter, a single trace conditioning session of 50 trials was carried out. On training trials a 100 ms air-puff (aversive, un-conditioned stimulus), was delivered 250 ms (trace interval) after the end of the tone. Pressurized air at 5 psi was passed through a nozzle of tip-diameter 1–1.5 mm positioned ~1–2 cm away and aimed at the mouse's left eye. Air-flow was switched on and off with a computer-controlled electronic, solenoid valve (EV mouse valve, Clippard, Cinncinati, OH, USA). Every fifth trial was a probe trial (on which

no puff was delivered) however as these amounted to only 10 probe trials per mouse, data from these trials have not been used for any specific analysis. Only one training session of 50 trials was carried out per mouse. Longer sessions were seen to cause blinking fatigue and discomfort to the mice. Additional sessions over multiple days were not possible due to the acute nature of the imaging preparation ('Materials and methods: Mice and surgical procedure'). All trained mice were imaged, irrespective of usability of imaging data, while training was carried out. Microscope scan-mirrors continued to oscillate during the inter-trial interval, to remove auditory cues signaling the beginning or end of a trial.

## Awake, two photon calcium imaging of area CA1 cells

Hippocampal area CA1 cells were imaged in parallel with eye-blink acquisition and training. We used a custom-built two-photon microscope, built around a Tsunami, Ti-Sapphire, 80 MHz pulsed laser (Spectra Physics, Mountain View, CA, USA) tuned to 810 nm for excitation with a water dipping objective lens (5CFI75 LWD 16x, NA 0.8, Nikon, Japan). Emitted fluorescent light was detected using an analog GaAsP PMT (H7422P-40; Hamamatsu, Japan). The amplified signal was binned over 2 µs pixel times. Imaging and training were carried out in complete darkness.

## Data analysis

All analysis was carried out using custom-written programs (Matlab, Mathworks, Natick, MA, USA), unless otherwise mentioned. Matlab code for the behavioral learning curves analysis (*Wirth et al., 2003*; *Smith et al., 2004*) and meta k-means analysis (*Ozden et al., 2008*; *Dombeck et al., 2009*) is available with the authors of the studies to first use it. Code for the main analyses carried out for this study is available for download at https://github.com/mehrabmodi/CA1-sequences-data-analysis-code.git.

### Behavioral analysis

Magnetometer trace baseline was calculated as the median of values recorded in a 2.4 s time window just prior to tone stimulus delivery. The entire trace was baseline-subtracted using this value. Blink response to tone was calculated by summing data values acquired from tone onset to air-puff onset (600 ms in total). Significant or conditioned blink responses (CR) were identified using a threshold given by,

$$\text{threshold} = \mu(\text{blink}_{\text{pre}}) + 2\sigma(\text{blink}_{\text{pre}}) \tag{1}$$

where $\text{blink}_{\text{pre}}$ is the area under the pre-training blink response curves in the time interval between tone onset and puff onset. Using this threshold, each trial was classified as either a blink (1) or a non-blink (0) trial, yielding a binary response vector for each mouse. CRs were identified in a 600 ms time-window after tone onset.

Hence, the probability of blink responses by chance was calculated as below.

$$p_{\text{chance}} = 1/(n_{\text{time windows}}) = 0.11 \tag{2}$$

Here, $p_{\text{chance}}$ is the probability of a well-timed CR by chance and $n_{\text{time windows}}$ is the number of 600 ms time windows before puff onset (5600/600 = 9.33). We then used a previously described expectation maximization algorithm to calculate learning curves for each mouse, as well as the learning trial if the mouse was identified as a learner (*Smith et al., 2004*). For the purposes of analysis, the mean 'learning-trial' across trained learners was assigned to all pseudo-conditioned mice. As an independent analysis of behavioral performance, we also calculated a performance score (PS) for each mouse. PS was defined as the ratio of CR frequency to spontaneous blink frequency. For the identification of spontaneous blinks, four 600 ms time-windows (2400 ms totally) before tone onset were considered. The same criterion for significant blinks was applied to these time windows and the fraction of windows (4 × 50 = 200 total windows) containing significant blinks was calculated as the spontaneous blink fraction.

### Image analysis

An averaged image from the first five frames of the first trial was used to manually draw regions of interest (ROIs) around cell-bodies. An averaged image from the first five frames of each trial was then used as a reference image for that trial. Each frame in the trial was aligned to this reference image using the TurboReg plugin for ImageJ (*Thevenaz et al., 1998*) to correct for movement of the imaged field during the trial. This registration method does not correct for within-frame motion. For each frame, cell intensity was calculated as the mean intensity of pixels within its ROI. Normalized, background-subtracted ΔF/F traces were calculated from these according to the equation,

$$\left(\Delta F \middle/ F\right)_i = F_i \middle/ F_{baseline} - 1 \qquad (3)$$

where $F_i$ is cell fluorescence on frame-$i$ and $F_{baseline}$ is the median fluorescence intensity of the cell on that trial. Only 4 s of data around the time of tone delivery were considered for this analysis, wherein the baseline was considered to be stable. Two metrics of cell activity during a 4 s long, spontaneous activity period (5 s away from any stimulus) were used to test for stability of imaging datasets. The summed area under the curve and the number of peaks detected were both calculated for the first and last 25% of trials. Datasets from mice showing statistically significant differences (two sample $t$ test, $p<0.05$) between early and late scores for both metrics were discarded. For all activity-timing related analyses, unless otherwise mentioned, ~50% of neurons were discarded as they had close to chance level timing-reliability scores ('Results: After learning, area CA1 cells show reliably timed, sequential calcium-responses').

## Peak time reliability score

The time of peak response (PT) for each cell was identified in averaged ΔF/F traces in the tone-onset to puff-onset period. Only every alternate trial, after the learning trial was used for this calculation (*Figure 3A*). The rest of the trials after learning trial were used to calculate the ridge to background ratio scores for calculating the reliability score. These trials were averaged and the summed area under PT and the two neighboring points was computed. The ratio of this area to that under all other points in the averaged trace was defined as the ridge to background ratio (*Figure 3C*). As a control condition, these traces were given random time-offsets and then averaged. An independent PT was identified for each random-offset, averaged trace and ridge to background ratio calculated for it (*Figure 3B,C*). This was repeated 5000 times for each cell's data and averaged. The reliability score was then calculated individually, for each cell, as the ratio of the ridge to background ratio for aligned traces to that of random-offset traces (*Figure 3D*). To capture trends in peak timing reliability, the same score was calculated for blocks of five trials each. For these and all other calculations, pseudo-conditioned animal data was aligned by time of delivery of the tone stimulus (*Figure 3F*).

## Time-decoder analysis

Only data from trials after the learning trial was considered. The decoder first built a template of average, expected population response vectors (prv) for each frame after tone-onset, up to the frame just prior to puff onset. The decoder then made time-predictions by comparing individual prvs from a non-overlapping test trial set to the template prvs. ΔF/F values from frames between tone onset and puff onset were taken for every alternate trial after the learning trial and averaged to obtain a set of expected prvs, corresponding to each frame in the period of interest. The prv for a single frame (current frame) from one of the remaining trials was taken. The dot product of the current frame prv with each of the expected prvs was then computed. The decoder's performance was scored by taking the dot product corresponding to the current frame number (correct frame) and dividing it by the summed dot products corresponding to in-correct frame numbers, weighted by distance from the correct frame. The correct frame dot product was given a weight equal to the total weight given to incorrect frame dot products. Incorrect frame dot product weights increased linearly with distance from correct frame, starting two frames away. As a control, dot-product vectors were randomly scrambled and performance score calculated. This was repeated 500 times and averaged to get the mean random-control performance score. In this manner, prediction scores were calculated for all single-frame prvs from all the remaining trials, not used to construct the expected population response template.

## Noise-correlation analysis

Spontaneous period correlations in activity were calculated for all neurons in a given dataset. A 4 s long stretch of the ΔF/F trace for each cell was taken from a period >4 s away from tone or puff delivery from each considered trial. The traces for a cell were stitched together for groups of trials and correlation coefficients were calculated for each pair of these, extended traces. In this way, noise correlations were computed for all appropriate cell pairs in each dataset. Trial-groups were considered as described in *Figure 4A, and 6A,B*. Note that for most analyses involving spontaneous activity correlations, cells were not discarded on the basis of their reliability scores as in the case of time-tuning analyses. The only exception is the analysis for *Figure 4—figure supplement 1B*. Here, neurons were classified into two groups, one with high activity timing reliability scores, and the other with low activity timing reliability

scores. Spontaneous activity correlations were then calculated for pairs of neurons within each group that shared the same activity peak timing.

## Clustering analysis

Meta k-means was employed to sort neurons into groups, so as to maximize within group correlations while minimizing across group correlations (*Ozden et al., 2008*). Briefly, the k means ++ algorithm (*Arthur and Vassilvitskii, 2007*) was initialized with k = 3 and was run 2000 times to generate clusters, after which cell-pairs falling into the same k-means ++ cluster for >80% of iterations were placed into 'meta-clusters'. We also initialized k-means ++ with k = 2 and k = 4, and similar results were obtained. These meta-clusters were then iteratively combined using pair-wise correlation coefficient as a criterion, as described previously (*Dombeck et al., 2009*) to yield the final clusters. Since the final clusters are formed by iterative pooling of meta-clusters, the number of clusters does not need to be pre-specified. The number of clusters obtained ranged from 2 to 8, while the k-value used to initialize k-means ++ in the reported data was 3.

## Grouping or cluster similarity score

The cluster similarity score was calculated for pairs of cell correlation clusters, with an aim to characterize how similar the two clusters were. First, the proportion of neurons belonging in common to both clusters was calculated, followed by calculating the overlap seen by chance, when random neurons were assigned to equal sized clusters. This is because overlap expected by chance depends on the numbers and sizes of clusters. The ratio of these two overlap measures was defined as the similarity score. In order to match clusters across trial sets, all possible combinations of cluster pairing in the two datasets were considered, and the one with the highest overlap proportion was used to calculate the similarity scores.

## Statistical testing with multiple comparisons

When making comparisons between more than two samples, we used the Tukey-Kramer honest significant difference test in conjunction with a one-way analysis of variance. The test calculates 99% confidence intervals around the mean of each sample, and reports significant differences between any pair of samples if the intervals do not overlap. The calculation of intervals accounts for the increased chances of type-1 errors due to multiple comparisons. Since 99% confidence intervals were calculated, the chance of error is smaller than 1%. Hence, p values were reported as <0.01.

# Acknowledgements

The authors thank Robert Cannon, Sachin Deshmukh, Niraj Dudani and Aditya Gilra for valuable inputs and discussions. The authors thank Dinu Albeanu for inputs and the generous gifts of head bars and head clamps.

# Additional information

### Funding

| Funder | Grant reference number | Author |
| --- | --- | --- |
| Wellcome Trust | NLO–056727/Z/99/B | Mehrab N Modi, Ashesh K Dhawale |
| Department of Biotechnology, Ministry of Science and Technology | BT/PR12531/BRB/10/747/2009 | Mehrab N Modi, Ashesh K Dhawale |
| National Centre for Biological Sciences | | Mehrab N Modi, Ashesh K Dhawale |

The funders had no role in study design, data collection and interpretation, or the decision to submit the work for publication.

### Author contributions

MNM, AKD, Conception and design, Acquisition of data, Analysis and interpretation of data, Drafting or revising the article; USB, Analysis and interpretation of data, Drafting or revising the article

## Ethics

Animal experimentation: All experimental procedures were approved by the National Centre for Biological Sciences institutional animal ethics committee (Protocol number: USB - 19 - ½011), in accordance with the guidelines of the Government of India (animal facility CPCSEA registration number: 109/1999/CPCSEA) and equivalent guidelines of the Society for Neuroscience. All surgery was performed under isoflurane anesthesia, and every effort was made to minimize suffering.

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
