## [Decision Letter]

Thank you for sending your work entitled “CA1 cell activity sequences emerge after reorganization of network correlation structure during associative learning” for consideration at *eLife*. Your article has been favorably evaluated by a Senior editor (Eve Marder) and 3 reviewers, one of whom is a member of our Board of Reviewing Editors.

The Reviewing editor and the reviewers discussed their comments before we reached this decision, and the Reviewing editor has assembled the following comments from the three reviews to help you prepare a revised submission.

This manuscript uses imaging to examine activity dynamics generated in area CA1 of the hippocampus while animals learn a conditioned eyeblink response. Since the US and CS are separated in time, they must be linked by neural activity in the intervening period. The authors find that relatively uncoordinated activity in CA1 reorganizes into a reliable sequential firing pattern that spans the time between CS and US, which they propose bridges this time interval. This paper offers two main contributions. First, observing these firing patterns during eyeblink conditioning expands the range of behavioral paradigms where sequences are observed, emphasizing their generality and hippocampal involvement in linking time-separated events. Second, the authors directly observe how sequential activity emerges over training. They find parallel changes in the structure of spontaneous correlations, implying changes in functional connectivity of the hippocampal network. While as the authors note, correlations do not definitively establish the nature of these changes, it does point towards redistributed functional interactions.

There were, however, several concerns and recommendations by the reviewers that should each be addressed in a revision:

1) The measure of learning – the trial on which the peak conditioned eyelid response is observed – is not intuitive and crude. It might be expected that the magnitude of the eyelid response might continue to grow long after learning, so a measure of when the first reliable CRs occur likely provides a more sensitive measure of initial learning. An even better approach would incorporate the sophisticated learning curve analysis used by Wirth et al. (Science, 2003) which tracks correct learning responses and would likely allow you to use eyeblink magnitude directly without setting an arbitrary threshold for CR or no-CR. This is viewed as important because you use the trial on which learning occurs to make conclusions about whether time cells precede or follow learning.

2) The reliability and temporal decoder scores are not intuitive. Reviewers had no idea whether a score of “3” is highly reliable or just statistically significant but a small effect. And it's not clear what statistical tests were used to generate p-values shown for these findings.

3) Why did the K-means analysis only test 2-4 clusters? Perhaps there are many?

4) It is not clear what the analyses of inter-trial-interval (ITI) correlations mean. Is this aimed to test whether time cell firing sequences are a “synfire” chain? How does correlation during the ITI imply anything about firing chains during the trace interval? A closer connection between connectivity and time cell sequences might be made if they were able to show that high ITI correlations occur only in strong time cells and not in other cells that do not exhibit temporally specific trace period firing.

5) A major strength of the paper is that it follows changes through training. These data could help address whether learning increases reliability of pre-existing responses already tied to specific time points (i.e., reinforces a pre-existing sequence), vs reconfigures activity by shifting peaks in time or generating new responses. The data point strongly to reconfiguration but some more in-depth analysis addressing this could be a nice addition. For example: Figure 3 shows an interesting single-cell example where firing occurs at similar times across trials, but more reliably after learning. The reliability score will tend to collapse differences in probability at the same time, vs changes in timing (or reliability of timing) of peaks, into a single number. It would be useful to address these separately – e.g., comparing the variability in peak times for each neuron across trials, before and after training and/or comparing the change in peak time. Does learning primarily affect mean timing of activity, variability of timing across trials, or reliability at a specific time point?

6) A central focus of the manuscript is changes in activity dynamics before and after learning. Figure 2 show clear strong peaks sequentially ordered across the CS-US interval. The heading of section 3 could be taken to mean that this structure is absent before learning. However, the data in Figure 3—figure supplement 1 shows clearly observable, albeit less complete, activity sequences. A side-by-side comparison of these data could help the reader evaluate the strength of learning-dependent effects. The reordering shown in supplemental panels A/B is a particularly striking learning outcome.

7) The authors have a substantial dataset of spontaneous activity, analyzed for noise correlations but not temporal structure. Depending on strength and timing of events, sorting by peak response could lead to apparent ordering of even randomly occurring activity. As a baseline, it would be useful to know the results of using the same sorting analysis for epochs of spontaneous data. Beyond this, it may even be possible to ask whether spontaneous activity gains any temporal structure resembling the stimulus-triggered sequences after training.

8) The authors describe the emergence of novel temporal sequences starting at the beginning of the CS (tone presentation) and evolving in time in the “learner” group and relate them to learning. For comparison, they use a control group that underwent a “pseudo-conditioning”. However, if one of the main goals is to investigate the neuronal dynamics underlying learning, the best control for the learner group should be the non-learner group that underwent the exact training protocol, but failed to learn the association. The authors should include the non-learner group in all of the relevant analyses and in the comparisons with the learner group. This could get them one step closer to understanding the neuronal correlates of learning.

9) The cellular activity was imaged at 11-16 Hz resulting in 70-90 ms-long frames. This is a quite large time interval for cellular physiology. How do the authors relate the long calcium transients to the spiking activity of the neurons occurring within such transients? Many possible different neuronal sequences can occur within each of the 7-8 frames composing the 600 ms interval between CS and US. For instance in Figure 2 frames 5 and 6 contain neurons 25-35 and 36-52, respectively which can fire in multiple different sequences within frames 5 and 6, across trials undetectably. Also, scale and units should be added to the x-axis (Time) in Figure 2.

10) Were the neurons that were active during the tone after training also active during the tone in the pre-training? How do we know the neuronal sequences in CA1 bear any meaning to this task? Does the non-learner group also exhibit neuronal sequences bridging the CS and US?

11) How do the authors explain that the noise correlations (NC) declined toward the end of the session? Were the animals maintaining equal attention to the task toward the end? The authors should also refer and relate to the trial by trial correlations in spiking activity of place cells in spatial environments that show steady increases with more training (e.g., [13], eLife; [7], Neuron).

12) In Figure 3, the Trace group shows average Reliability Score values around 3. In Figure 3, the same variable ranges from a minimum of ∼1 in early trials to a maximum of ∼1.4. The numbers in 3D and 3F don't match. Please explain the difference.

13) What is being displayed in Figure 5? Please explain in more detail. In Figure 7, spikes should not go in time beyond the time of air puff delivery as they could by emitted in response to the air puff. In Figure 7, late in the training session (bottom), if increased correlations should be considered a mark of learning, why do they decline with more training? Please explain. Figure 6–figure supplement 1; why are NC values in trace conditioning group of mice that fail to learn in the beginning as high as the maximum NC values of learners later during training? Figure 3; color coding and labeling is not entirely clear. Are these different image frames at which the cell responded?

Also panels D and E in Figure 3—figure supplement 1 have no legend. These appear to show the number of cells active for each time epoch before learning. Again a direct side-by-side comparison of this data before and after learning could be valuable.

---

## [Author Response]

*1) The measure of learning – the trial on which the peak conditioned eyelid response is observed – is not intuitive and crude. It might be expected that the magnitude of the eyelid response might continue to grow long after learning, so a measure of when the first reliable CRs occur likely provides a more sensitive measure of initial learning. An even better approach would incorporate the sophisticated learning curve analysis used by Wirth et al. (Science, 2003) which tracks correct learning responses and would likely allow you to use eyeblink magnitude directly without setting an arbitrary threshold for CR or no-CR. This is viewed as important because you use the trial on which learning occurs to make conclusions about whether time cells precede or follow learning*.

We thank the reviewers for the helpful comment and for recommending a previously used algorithm to help address the issue. The reviewers correctly pointed out that using the peak of the moving-window averaged behavioral response curve as the trial where learning occurred would be ill-defined and arbitrary. In the study recommended by the reviewers (50), and a subsequent article with many of the same authors (43), we found an analysis algorithm that allowed us to address the problem of rigorously identifying the learning trial for each mouse. We describe the use of this algorithm in the revised manuscript.

We next re-analyzed and re-plotted all our data that depended on the learning trials of mice and found that all the conclusions in our original submission remain un-changed (Figure 3 and Figure 4—figure supplement 1).

The reviewers additionally state that our classification of individual trials as response or failure depends on an ‘arbitrary’ threshold. We would like to clarify that the threshold for blink classification is based on a two standard deviation criterion, where we compare blink responses during training to the pre-training blink responses to the tone stimulus. The threshold defined in our original manuscript (Materials and methods – Data analysis) and used in the revised submission as well is given by the equation below:

threshold=µ(blinkpre)+2s(blinkpre)

where µ(blink_pre_) is the mean size of the blink response to the tone, prior to training and σ(blink_pre_) is the standard deviation of pre-training blink response size. In fact, in the study recommended by the reviewers (50), the authors first obtain binary response vectors and then use the algorithm described above. This is the methodology we have adopted as well in this revision.

*2) The reliability and temporal decoder scores are not intuitive. Reviewers had no idea whether a score of “3” is highly reliable or just statistically significant but a small effect*.

In the revised manuscript, we have expanded on the reliability and temporal decoder scores. A new plot (Figure 3—figure supplement 1) deals with the principle underlying the reliability score. We have also related the reliability score to experimental observations reproduced in the manuscript to make it more intuitive. As explained in the revised manuscript, in Results: “After learning, area CA1 cells show reliably timed, sequential calcium-responses”.

With regard to the time decoder scores, we have expanded the Results section of the revised manuscript.

*And it's not clear what statistical tests were used to generate p-values shown for these findings*.

We thank the reviewers for indicating that this point was not clear. We now discuss it in the Materials and methods section of the revised manuscript.

*3) Why did the K-means analysis only test 2-4 clusters? Perhaps there are many*?

The reviewers have raised a concern regarding the forcing of a clustering algorithm to allot cells into a pre-defined number of clusters. However, we would like to point out that the meta k-means clustering algorithm used in our study does not need the user to pre-specify the number of clusters that cells are allotted to (41; 9). The algorithm relies on an initial step of repeated clustering by the k-means++ algorithm with pre-specified numbers of clusters (step 1 in Figure 8). This is the k value range of 2-4 referred to by the reviewers. After 2000 randomly initialized runs of the k-means++ algorithm, we get many different sets of clusters. The meta k-means algorithm then identifies cells that are co-clustered in more than 80% of k-means ++ runs and groups them into meta clusters initially. These meta clusters are then iteratively merged, based on the similarity of their averaged, activity traces, to yield a variable, unspecified number of final clusters, of un-specified sizes. The average number of correlation clusters we obtained using this algorithm ranged from 2 to 8, whereas the k-value used to initialize k-means++ was 3. Furthermore, we would like to point out that the central finding we report does not depend on the number of clusters per se, but rather on their relative spatial organization and whether or not they change during learning (Figures 5 and 6 in the revised manuscript).Author response image 1.Schematic describing how the meta k-means clustering algorithm works.

For the purposes of clarity, we have included here a schematic diagram describing how the algorithm works (Figure 8). This diagram explains how the number of clusters is not pre-determined. To highlight this fact, we have also expanded the Materials and methods section of the revised manuscript.

*4) It is not clear what the analyses of inter-trial-interval (ITI) correlations mean. Is this aimed to test whether time cell firing sequences are a “synfire” chain? How does correlation during the ITI imply anything about firing chains during the trace interval? A closer connection between connectivity and time cell sequences might be made if they were able to show that high ITI correlations occur only in strong time cells and not in other cells that do not exhibit temporally specific trace period firing*.

We thank the reviewers for pointing out a potential source of confusion. As mentioned in the original manuscript, in Results (Noise-correlations increase transiently during learning, especially between similarly time-tuned neurons) and the Discussion, we have used changes in spontaneous activity correlations as a proxy for changes in network connectivity. We reported that when considering only neurons with high reliability scores, neurons sharing the same time-tuning showed higher spontaneous activity correlations than random neuron-pairs (Results; Figure 4). As described in the Discussion of the original manuscript, CA1 neuron synfire chains are unlikely, given the extremely low CA1-CA1 connection probability (Andersen, Morris, et al., 2007). Instead, as described, we feel that the altered spontaneous correlations reflect changes in upstream, CA3-CA3 or CA3-CA1 connectivity. We have additionally done the analysis recommended by the reviewers, and included it in Figure 4—figure supplement 1. This figure is discussed in the revised manuscript.

*5) A major strength of the paper is that it follows changes through training. These data could help address whether learning increases reliability of pre-existing responses already tied to specific time points (i.e., reinforces a pre-existing sequence), vs reconfigures activity by shifting peaks in time or generating new responses. The data point strongly to reconfiguration but some more in-depth analysis addressing this could be a nice addition. For example:*
Figure 3
*shows an interesting single-cell example where firing occurs at similar times across trials, but more reliably after learning. The reliability score will tend to collapse differences in probability at the same time, vs changes in timing (or reliability of timing) of peaks, into a single number. It would be useful to address these separately – e.g., comparing the variability in peak times for each neuron across trials, before and after training and/or comparing the change in peak time. Does learning primarily affect mean timing of activity, variability of timing across trials, or reliability at a specific time point*?

We thank the reviewers for their suggestions on analysis. It is correctly pointed out that the reliability score for CA1 cell activity peak timing in the original manuscript does not distinguish between two possible phenomena. A cell may have a fixed activity peak timing from the beginning of the session, but not fire on every trial. In such a case, if the cell gradually started firing with higher probability at the same, fixed peak-timing, its reliability score would increase (Figure 9).. As pointed out by the reviewers, the cell in Figure 3 is one such example. Alternatively, the proportion of trials on which the cell fires may not change with training. Instead, the timing of peak activity itself might change (Figure 9) and such a cell would show an increase in reliability score as well.Author response image 2.Schematic diagram depicting two possible patterns of learning related changes in timing of activity peaks. **A** Activity peak timing is fixed from the beginning of the session, only reliability of trial to trial activity increases with learning. **B** Activity peak timing changes with training, with reliable activity at a fixed timing being seen only towards the end of the session.

As already demonstrated in the original manuscript, there is extensive re-distribution of activity peak- timings with learning. The distribution of peak timings is heavily skewed towards the tone period in the pre-training, tone alone session (Figure 3 in the revised manuscript). Early in the training session, with the introduction of the air-puff stimulus, this distribution shifts markedly, with a peak near the time of puff presentation (Figure 3 in the revised manuscript). This extensive re-organization of activity peak timings is also evident in Figure 2. Here, averaged traces from the pre-training, tone alone session have been sorted as per the cells’ peak-timings after learning. The lack of sequentially timed peaks with this ordering indicates that peak timings change during learning.

We address this issue in the revised manuscript.

*6) A central focus of the manuscript is changes in activity dynamics before and after learning.*
Figure 2
*show clear strong peaks sequentially ordered across the CS-US interval. The heading of section 3 could be taken to mean that this structure is absent before learning. However, the data in*
Figure 3—figure supplement 1
*shows clearly observable, albeit less complete, activity sequences. A side-by-side comparison of these data could help the reader evaluate the strength of learning-dependent effects. The reordering shown in supplemental panels A/B is a particularly striking learning outcome*.

The figure panels Figure 3—figure supplement 1 from the old version of the manuscript have now been moved to Figure 2 as recommended by the reviewers (Figure 2 in the revised manuscript) to allow side by side comparison.

*7) The authors have a substantial dataset of spontaneous activity, analyzed for noise correlations but not temporal structure. Depending on strength and timing of events, sorting by peak response could lead to apparent ordering of even randomly occurring activity. As a baseline, it would be useful to know the results of using the same sorting analysis for epochs of spontaneous data*.

The reviewers have pointed out the possibility that the sorting analysis used to determine activity peak timings could lead to apparent ordering of even randomly occurring activity. The baseline test suggested is to use the same analysis with data from the spontaneous activity periods, rather than from the stimulus presentation periods. We would like to clarify that this analysis had already been done in the originally submitted manuscript. Figure 3 all have spontaneous period activity analyzed in the same manner, to serve as a control. In all these cases, stimulus period data from trace learners showed significantly higher scores for sequential activity than spontaneous period data.

*Beyond this, it may even be possible to ask whether spontaneous activity gains any temporal structure resembling the stimulus-triggered sequences after training*.

The reviewers make the interesting suggestion that there might be sequence replay events during spontaneous period activity. To check for this, we performed additional calculations, the results of which are plotted in Figure 4—figure supplement 1. This is discussed in the revised manuscript.

*8) The authors describe the emergence of novel temporal sequences starting at the beginning of the CS (tone presentation) and evolving in time in the “learner” group and relate them to learning. For comparison, they use a control group that underwent a “pseudo-conditioning”. However, if one of the main goals is to investigate the neuronal dynamics underlying learning, the best control for the learner group should be the non-learner group that underwent the exact training protocol, but failed to learn the association. The authors should include the non-learner group in all of the relevant analyses and in the comparisons with the learner group. This could get them one step closer to understanding the neuronal correlates of learning*.

We appreciate the utility of direct comparisons between learners and non-learners. We discuss this in our originally submitted manuscript, and now in the revised manuscript. We would like to re-iterate that non-learners likely consist of a heterogeneous group of mice, at various stages of learning and thus, with different extents of learning-related changes in multiple brain regions. Hence, pooling CA1 activity data from such a heterogeneous group is likely to be confounded with conflicting data from each mouse. However, for the sake of completeness, we had included data from non-learners in all major analyses and placed these plots in supplementary figures. Out of deference to the reviewers’ recommendation, these panels have now been shifted to the main text (Figures 3, 4 and 6 in the revised manuscript). Non-learner data has also been included in some new figures (Figure 3—figure supplement 1).

*9) The cellular activity was imaged at 11-16 Hz resulting in 70-90 ms-long frames. This is a quite large time interval for cellular physiology. How do the authors relate the long calcium transients to the spiking activity of the neurons occurring within such transients? Many possible different neuronal sequences can occur within each of the 7-8 frames composing the 600 ms interval between CS and US. For instance in*
Figure 2
*frames 5 and 6 contain neurons 25-35 and 36-52, respectively which can fire in multiple different sequences within frames 5 and 6, across trials undetectably. Also, scale and units should be added to the x-axis (Time) in*
Figure 2.

The reviewers have correctly pointed out that the time-resolution of our calcium-imaging read-out of neuronal activity does not allow us to capture all possible neurophysiological phenomena. We discuss this in the Discussion section of the revised manuscript. Hence, we maintain that the claims made in our study are not affected by this caveat. A time label has been added to the time-scale bar in Figure 2.

*10) Were the neurons that were active during the tone after training also active during the tone in the pre-training*?

48% of cells with high reliability scores were active during the tone period in the pre-training, tone only session. Post-learning, while 40.0 % of cells had peak times during tone presentation, only 19.2 % of high reliability cells were tone-responsive both prior to and post learning. This is indicative of extensive changes in activity timing due to learning. See the detailed response to reviewer comment 5. There, we discuss analyses that show that the activity peak timings for neurons change during training.

*How do we know the neuronal sequences in CA1 bear any meaning to this task? Does the non-learner group also exhibit neuronal sequences bridging the CS and US*?

It is only the learner group who form reliably timed activity sequences. As shown in the originally submitted manuscript, trace learners have higher reliability scores for sequentially timed activity (Figure 3 in the revised manuscript), as well as higher time-decoder prediction scores (Figure 3 in the revised manuscript) than pseudo-conditioned mice as well as non-learner mice that failed to learn the association. Hence, we concluded that the emergence of sequential activity is related to learning the behavioral task.

*11) How do the authors explain that the noise correlations (NC) declined toward the end of the session? Were the animals maintaining equal attention to the task toward the end*?

The reviewers have highlighted an interesting phenomenon. As learning progressed, we observed an elevation in cell-cell spontaneous activity correlations (Figure 4 in the revised manuscript). However, these correlations went back down towards the end of the session.

We explain this in the revised manuscript.

*The authors should also refer and relate to the trial by trial correlations in spiking activity of place cells in spatial environments that show steady increases with more training (e.g.,*
[13]*, eLife;*
[7]*, Neuron)*.

The reviewers have cited two studies of high relevance to ours that we failed to cite in the original manuscript. We thank the reviewers for highlighting this accidental omission on our part. Both studies have been cited and discussed in the revised version of the manuscript.

*12) In*
Figure 3*, the Trace group shows average Reliability Score values around 3. In*
Figure 3*, the same variable ranges from a minimum of ∼1 in early trials to a maximum of ∼1.4. The numbers in 3D and 3F don't match. Please explain the difference*.

The reviewers have correctly pointed out that the cell-averaged reliability score in Figure 3 is much higher than that seen in Figure 3 (3 E in the revised manuscript, hereafter referred to as Figure 3). We address the difference in scores in the Results section of the revised manuscript.

*13) What is being displayed in*
Figure 5*? Please explain in more detail*.

We thank the reviewers for pointing out that the purpose of Figure 5 is unclear. Figure 5 illustrates spontaneous activity correlation clusters (Results – Correlated cell-groups are organized into spatial clusters that are re-organized by training). Panel A shows the ΔF/F, spontaneous activity traces for all the cells in a dataset, sorted according to the correlation clusters they belong to. It shows how cells within a cluster display highly correlated activity that is different from that of cells in other clusters. We have added an inset to the original figure where cell responses are more clearly visible. We have also inserted a clearer explanation of the motivation for its inclusion in this figure’s legend (Figure 5 in the revised manuscript).

*In*
Figure 7*, spikes should not go in time beyond the time of air puff delivery as they could by emitted in response to the air puff*.

The reviewers have correctly pointed out that the schematic diagram in Figure 7 (middle and bottom panels) should not have cells that spike during the period of air puff delivery. This was an un-noticed error on our part. The schematic has been edited to ensure that none of the cells do so.

*In*
Figure 7*, late in the training session (bottom), if increased correlations should be considered a mark of learning, why do they decline with more training? Please explain*.

See the detailed response to reviewer comment 11. There, we discuss other studies that make similar observations and how we interpret the data and its implications for hippocampus-dependent learning.

*Figure 6–figure supplement 1; why are NC values in trace conditioning group of mice that fail to learn in the beginning as high as the maximum NC values of learners later during training*?

The mean noise correlations for non-learners in the first trial bin (mean noise correlation (NC) 0.23) is close to, but slightly lower than the peak value reached in the case of learners (mean NC 0.24). We speculate that the initially elevated noise correlations are a signature of a pre-existing network state that is detrimental to the rapid acquisition of trace conditioning.

We ruled out two possible, trivial reasons for the high NC in the first trial bin. First, we showed that the elevated NC is not due to a few, high outliers skewing the reported mean. The box and whisker plot in Figure 10 depicts the same data as that plotted for the noise correlations curve for non-learners (cyan curve) in Figure 4 of the revised manuscript. The box and whisker plot shows that the higher mean seen in the first trial bin is not due to outliers, but is in fact, a result of most of the correlation values for that bin being higher. The upper quartile, as well as the highest data value within 1.5 quartiles of the upper quartile, are far higher for the first bin than for subsequent bins. Second, we ruled out the trivial possibility that what we measure as noise-correlations using data acquired 4 seconds after puff stimulus delivery are in fact, correlations in long-sustained responses to the puff stimulus. We calculated peak response widths for each cell in the same manner as for Figure 2 in the revised manuscript. To look for sustained activity after the puff stimulus during trials 1 to 5 for non-learners, we calculated response widths in the 4 s period immediately after the puff for trials 1 to 5. We found that no cell’s response width was longer than 1 s. This rules out the possibility of sustained, puff stimulus responses 4 s later, during the segment of data used to calculate noise correlations.Author response image 3.**A** Mean noise correlations for non-learners are high in the first trial bin. This is not due to the presence of a few outliers. This panel is a box and whisker plot for the noise correlation data corresponding to the non-learners curve (cyan curve) in Figure 4. The boxes depict upper and lower quartiles around the median (black circle) noise correlation values for each window of trials. The whiskers represent the most extreme data values to lie within 1.5 quartiles from the upper and lower quartiles. **B** Response widths during the post-stimulus period are shorter than 1 s for cells from non-learners. Cell dF/F traces for the first 5 trials were averaged and then response widths were calculated for these trial-averaged traces for a 4 s period after the end of the puff stimulus. This was done in the same manner as for Figure 2.

Figure 3*; color coding and labeling is not entirely clear. Are these different image frames at which the cell responded*?

We thank the reviewers for pointing out the omission of the title of the color bar. This title has been inserted into Figure 3 (Figure 3—figure supplement 1 in the revised manuscript). Additionally, the words ‘frame number’ have been added to the figure’s legend to clarify that cell-masks are in fact color-coded by the frame number of the peak calcium response or peak timing.

*Also panels D and E in*
Figure 3—figure supplement 1
*have no legend. These appear to show the number of cells active for each time epoch before learning. Again a direct side-by-side comparison of this data before and after learning could be valuable*.

We thank the reviewers for pointing out this accidental omission on our part. The necessary legends have now been included in the revised submission. Also, the panels have been moved to Figure 3, as recommended by the reviewers (Figure 3 in the revised manuscript). The reviewers’ interpretation of what these panels depict was correct.

References

1. Wirth S, Yanike M, Frank LM, Smith AC, Brown EN, Suzuki WA. 2003. Single Neurons in the Monkey Hippocampus and Learning of New Associations. *Science*
**300**:1578–81. doi: 10.1126/science.1084324.

2. Smith AC, Frank LM, Wirth S, Yanike M, Hu D, Kubota Y, et al. 2004. Dynamic Analysis of Learning in Behavioral Experiments. *J Neurosci*
**24**:447–61. doi: 10.1523/JNEUROSCI.2908-03.2004.

3. Ozden I, Lee HM, Sullivan MR, Wang SS-H. 2008. Identification and Clustering of Event Patterns From In Vivo Multiphoton Optical Recordings of Neuronal Ensembles. *J Neurophysiol*
**100**:495–503. doi: 10.1152/jn.01310.2007.

4. Dombeck DA, Graziano MS, Tank DW. 2009. Functional Clustering of Neurons in Motor Cortex Determined by Cellular Resolution Imaging in Awake Behaving Mice. *J Neurosci*
**29**:13751–60. doi: 10.1523/JNEUROSCI.2985-09.2009.

5. Andersen P, Morris R, Amaral D, Bliss T, O’Keefe J. 2007. The Hippocampus Book [Internet]. Available from: http://books.google.co.in/books?id=zg6oyF1DziQC

6. Lee AK, Wilson MA. 2002. Memory of Sequential Experience in the Hippocampus during Slow Wave Sleep. *Neuron*
**36**:1183–94. doi: 10.1016/S0896-6273(02)01096-6.

7. Cheng S, Frank LM. 2008. New Experiences Enhance Coordinated Neural Activity in the Hippocampus. *Neuron*
**57**:303–13. doi: 10.1016/j.neuron.2007.11.035.